# Effects of bryophyte and lichen cover on permafrost soil temperature at large scale

P. Porada[1,2], A. Ekici[3,4], and C. Beer[1,2]

[1]Department of Environmental Science and Analytical Chemistry, Stockholm University, Svante Arrhenius väg 8, SE-11418 Stockholm
[2]Bolin Centre for Climate Research, Stockholm University, 10691 Stockholm
[3]Earth System Sciences, Laver Building, University of Exeter, Exeter, UK
[4]Uni Research Climate, Bjerknes Centre for Climate Research, Bergen, Norway

*Correspondence to:* Philipp Porada (Philipp.Porada@aces.su.se)

**Abstract.** Bryophyte and lichen cover on the forest floor at high latitudes exerts an insulating effect on the ground. In this way, the cover decreases mean annual soil temperature and can protect permafrost soil. Climate change, however, may change bryophyte and lichen cover, with effects on the permafrost state and related carbon balance. It is therefore crucial to predict how the bryophyte and lichen cover will react to environmental change at the global scale. To date, current global land surface models contain only empirical representations of the bryophyte and lichen cover, which makes it impractical to predict the future state and function of bryophytes and lichens. For this reason, we integrate a process-based model of bryophyte and lichen growth into the global land surface model JSBACH. The model simulates bryophyte and lichen cover on upland sites. Wetlands are not included. We take into account the dynamic nature of the thermal properties of the bryophyte and lichen cover and their relation to environmental factors. Subsequently, we compare simulations with and without bryophyte and lichen cover to quantify the insulating effect of the organisms on the soil.

We find an average cooling effect of the bryophyte and lichen cover of 2.7 K on temperature in the topsoil for the region north of 50° N under current climate. Locally, a cooling of up to 5.7 K may be reached. Moreover, we show that using a simple, empirical representation of the bryophyte and lichen cover without dynamic properties only results in an average cooling of around 0.5 K. This suggests that a) bryophytes and lichens have a significant impact on soil temperature in high-latitude ecosystems and b) a process-based description of their thermal properties is necessary for a realistic representation of the cooling effect. The advanced land surface scheme including a dynamic bryophyte and lichen model will be the basis for an improved future projection of land-atmosphere heat and carbon exchange.

## 1  Introduction

Vegetation cover on the ground, consisting of bryophytes (mosses, liverworts, hornworts) and lichens, may play a significant role for the energy and carbon balance of high-latitude ecosystems, since it reduces the exchange of heat between atmosphere and soil (Soudzilovskaia et al., 2013). This insulating effect causes a decrease in soil temperature and thereby protects permafrost (Jorgenson et al., 2010; Turetsky et al., 2010). Consequently, a reduction in the surface coverage of bryophytes and lichens may result in increased soil temperature accompanied by thawing of permafrost (Rinke et al., 2008). This might lead to an increased release of carbon from the soil to the atmosphere which would accelerate global warming (Schuur et al., 2015),

although this may be partially compensated by increased productivity of e.g. shrubs in northern ecosystems. It is therefore crucial to determine if the bryophyte and lichen cover will expand or recede in response to a warming climate and elevated atmospheric $CO_2$. Depending on the direction of this response, the effect of bryophytes and lichens at high latitudes on global climate might represent either a positive or a negative feedback.

As a first step to analyse this feedback, we focus in this study on the effect of the bryophyte and lichen cover on soil temperature at high-latitudes. At the local to regional scale, experimental as well as modelling studies have assessed the effect of the bryophyte and lichen cover on soil temperature. Blok et al. (2011) show that the experimental removal of moss leads to increased ground heat flux and also increased soil evaporation. Gornall et al. (2007) observe higher soil temperatures in summer with decreasing thickness of the moss cover on the ground and also lower soil temperatures in winter. In a modelling

study, Bonan (1991) perform simulations with a local energy balance model for 20 forest stands in Central Alaska to estimate the effect of moss removal on soil temperature. They find an increase in soil temperature due to the reduction in moss ground cover. Jorgenson et al. (2010) use a local soil temperature model to quantify the effect of several factors on permafrost and find that vegetation on ground strongly promotes permafrost stability.

The insulating effect of the bryophyte and lichen ground cover on soil temperature is strongly modulated by bryophyte

and lichen moisture content. The thermal conductivity of the cover increases by around one order of magnitude from dry to water saturated bryophytes and lichens, while the heat capacity even increases by around two orders of magnitude (O'Donnell et al., 2009; Soudzilovskaia et al., 2013). Moreover, thermal conductivity increases strongly during the transition from liquid to frozen water inside the bryophyte and lichen cover, while heat capacity is reduced (Hinzman et al., 1991). Therefore, a large and important seasonality of the insulating effect is expected: In winter and, to a lesser extent, in autumn and spring, the

insulating effect of bryophytes and lichens will be small due to the high thermal conductivity of ice and water. In summer, however, a dry bryophyte and lichen cover may strongly insulate the ground, leading to a net cooling effect for the whole year. The relation between thermal properties of bryophytes and lichens and their water saturation is very similar for a large range of different species (Soudzilovskaia et al., 2013). However, the species show considerable differences in water holding capacity and other properties which affect their water status. Therefore, different species may exhibit differing water saturation under

similar climatic conditions.

Several global land surface models which attempt to simulate permafrost soil at high latitudes include a bryophyte and lichen ground cover, while others approximate near-surface vegetation by grass (e.g. Zhu et al. (2015)). Alternatively, the insulating effect is attributed solely to the carbon-rich uppermost soil layer (Koven et al., 2009). However, the dynamic nature of bryophyte and lichen surface coverage and the dependence of the insulating effect on moisture content have not been explicitly considered

so far in global land surface models. Beringer et al. (2001), for instance, extend the land surface model NCAR-LSM by lichen or moss topsoil parameterisations for an arctic tundra site, but they assume constant thermal conductivity and surface coverage of the organic layer. Ekici et al. (2014) use the global land surface model JSBACH to simulate permafrost at high latitudes and thereby include an organic layer above the soil which reduces heat exchange with the atmosphere. The layer is assumed to cover the whole land surface uniformly and is set to constant thermal properties. Recently, (Chadburn et al., 2015a) added

a representation of moss to the land surface model JULES. While the surface coverage of the moss layer is prescribed by a

static map, the thermal conductivity of the moss layer is dynamic. Instead of being related to moss water content, however, the thermal conductivity is determined by the moisture content of the uppermost soil layer. Subsequently, Chadburn et al. (2015b) extend JULES by an empirical scheme which determines moss "health" as a function of climate variables and consequently allows for the computation of a dynamic moss ground cover. In summary, the use of constant thermal properties and empirical relations in current global land surface models makes it difficult to predict the impact of future bryophyte and lichen functions on permafrost ecosystems.

Here we present a process-based model which simulates productivity and dynamic surface coverage of the bryophyte and lichen ground cover at the global scale. The model represents the organisms' water and ice content and thereby accounts for dynamic thermal properties of the cover. The basis for this model is the stand-alone dynamic non-vascular vegetation model LiBry which has been successfully applied to quantify global productivity by bryophytes and lichens (Porada et al., 2013) as well as estimating their contribution to global biogeochemical cycles (Porada et al., 2014). LiBry is fully integrated into the land surface scheme JSBACH of the Max Planck Institute Earth System Model. JSBACH simulates the carbon balance of vascular plants and the soil at the global scale, it includes a water and energy balance and also a representation of permafrost. Based on the thermal properties of the near-surface vegetation layer we use JSBACH to quantify the effect of bryophytes and lichens on heat transfer between atmosphere and soil and, therefore, on soil temperature. Since JSBACH does not include a scheme for wetland hydrology, the formation of peatlands cannot be simulated. Hence, LiBry in JSBACH mainly represents bryophyte and lichen growth on upland forest floor sites.

We compare the soil thermal regime of a JSBACH simulation including a dynamic bryophyte and lichen layer to a simulation where the thermal properties and surface coverage of this layer are set to constant and also to another simulation with the bryophyte and lichen cover switched off. In this way, we can assess quantitatively the impact of the bryophyte and lichen ground cover on soil temperature at high latitudes. In a next step, our process-based approach can be used to predict the role of bryophytes and lichens for high-latitude ecosystems under climatic change.

## 2 Methods

### 2.1 Model description

For our analysis we use the global land surface model JSBACH 3.0 (Raddatz et al., 2007; Brovkin et al., 2009), which is a part of the Max Planck Institute Earth System Model (MPI-ESM 1.1). JSBACH uses a process-based approach to simulate both physical and biochemical ecosystem functions, such as the exchange of energy and water at the land surface and carbon fluxes between atmosphere, vegetation and soil which are determined by photosynthesis and respiration. The soil part of JSBACH includes a soil carbon model (Goll et al., 2015) and a five-layer soil hydrology scheme (Hagemann and Stacke, 2015). It has been extended by Ekici et al. (2014) to take into account both several dynamic snow layers and the latent heat of fusion associated with freezing and thawing, which is the basis for simulating permafrost extent and active layer thickness.

To represent bryophyte and lichen ground cover, the process-based non-vascular vegetation model LiBry, described in detail by Porada et al. (2013), is integrated into JSBACH. LiBry is a stand-alone dynamic global vegetation model that uses climate

data, such as radiation, temperature and precipitation to predict photosynthesis, respiration and growth. The model combines approaches used in many dynamic vegetation models, such as the Farquhar photosynthesis scheme, with lichen- and bryophyte-specific processes, e.g. the adaptation of the metabolism to changing levels of water content.

LiBry uses the Farquhar scheme (Farquhar and von Caemmerer, 1982) to calculate photosynthesis. Thus, both $CO_2$ and photosynthetically active radiation (PAR) are limiting factors. The availability of PAR on the ground depends on the shading by trees and it is consequently inversely related to the leaf area index (LAI) of the vegetation above the bryophyte and lichen cover. In JSBACH, the fraction of PAR available at the ground can be directly computed from the simulated LAI of the overlying vegetation (Knorr, 2000) and it is then used as an input for LiBry. To estimate the availability of $CO_2$ for bryophytes and lichens, the model computes diffusion of $CO_2$ from the atmosphere into the organisms. The $CO_2$ diffusivity is decreasing with increased water content of the organisms due to narrowing of diffusion pathways and formation of water films (Cowan et al., 1992). In addition to $CO_2$ and PAR, the rate of modelled photosynthesis depends on surface temperature, which is calculated by JSBACH from the surface energy balance. Furthermore, the photosynthesis rate is related to the level of metabolic activity of bryophytes and lichens, which is controlled by their water content and which ranges from inactive for dry organisms to fully active for water saturated organisms. This is an important function for predicting the response to climate change as precipitation patterns are projected to change (Pachauri et al., 2014).

The water content of LiBry in JSBACH is computed from the balance of water input and loss at a given water storage capacity. For bryophytes and lichens, water input corresponds to rainfall or snowmelt and water loss takes place in form of evaporation. While JSBACH provides fluxes of rainfall and snowmelt, it does not compute the evaporation flux from ground based vegetation. In JSBACH, evapotranspiration is partitioned only into transpiration from vegetation, bare soil evaporation and evaporation from the interception reservoir. The latter includes water evaporating freely from vegetation surfaces in the canopy and on the ground. Since it is impractical to modify this scheme, we approximate evaporation from bryophytes and lichens by evaporation from the interception reservoir. This potentially neglects the morphological control of the organisms on evaporation. However, free evaporation is more suitable than transpiration by vascular plants to describe water loss of bryophytes and lichens (Nash III, 1996; Proctor, 2000). Hence, the water balance of the simulated bryophytes and lichens is coupled to the interception reservoir in JSBACH. This is done by adding the water storage capacity of bryophytes and lichens to the size of the interception reservoir and setting the water saturation of the bryophytes and lichens equal to the saturation of the interception reservoir.

In addition to processes involving liquid water, freezing and melting of water are taken into account in JSBACH. Consequently, water inside the bryophyte and lichen cover may be partly or completely frozen. Since frozen water occupies pore space, the size of the interception reservoir is reduced by the amount of ice inside the bryophyte and lichen cover as long as ice occurs.

The net primary productivity (NPP) of bryophytes and lichens is obtained in LiBry by subtracting respiration from gross photosynthesis, where respiration is simulated as a function of temperature and metabolic activity. Turnover of biomass associated with mortality is also considered, and it is modelled as a function of the protein content of biomass. The net growth of bryophytes and lichens is then determined by the balance of NPP and turnover. By multiplying net growth of simulated

bryophytes and lichens with their specific area, the growth in surface coverage is derived in the model. Without processes that reduce the bryophyte and lichen cover, the surface coverage would increase to 100 % in all areas where net growth is larger than zero in steady-state. Hence, a disturbance cycle is included into LiBry which periodically sets back the surface coverage to a small initial value. The surface coverage in steady-state is then obtained by averaging over a whole disturbance cycle.

LiBry in JSBACH is designed to predict the dynamics of the cover in transient scenarios of climate change. Consequently, the steady-state calculation of the surface coverage from the original LiBry is replaced by a dynamic scheme. This also accounts for potential changes in disturbance frequency. To avoid unrealistic fluctuations of the bryophyte and lichen cover, the dynamic disturbance scheme of LiBry in JSBACH accumulates losses of cover over a month. Surface coverage is then updated on a monthly basis: The simulated bryophyte and lichen cover is reduced by the accumulated losses due to disturbance, such as fire,

and it is increased by positive net growth accumulated over a month. Net growth is translated into coverage via the specific area of the organisms. Negative net growth is subtracted from the cover. The dynamic scheme also accounts in a simple way for dispersal and establishment of bryophytes and lichens (e.g. Bohn et al. (2011)): The absolute increase in cover per area of ground is both limited by the already existing cover, which generates the new biomass, as well as the free area that is still available for the growth of new cover. The transition of new cover from the existing cover to the free area may take place in

form of spores or via vegetative growth (Grime et al., 1990; Rogers, 1990). The efficiency of these processes is summarised in the dynamic scheme by an "expansion efficiency" (see Fig. 1). Equation 1 shows the change of bryophyte and lichen cover $\Delta A / \Delta t$ in one month:

$$\Delta A / \Delta t = E - R_{\mathrm{G}} - R_{\mathrm{D}} \tag{1}$$

where $E$ is the expansion of the bryophyte and lichen cover due to positive net growth, which means that accumulated gross

photosynthesis exceeds accumulated respiration and biomass turnover. $R_{\mathrm{G}}$ is the reduction of cover due to negative net growth and $R_{\mathrm{D}}$ stands for reduction of cover by disturbance. Expansion of cover $E$ is calculated as:

$$E = \begin{cases} min\left(GA(1.0 - A)\eta_{\mathrm{E}}, 1.0 - A\right) & \text{if } G \geq 0 \\ 0 & \text{if } G < 0 \end{cases} \tag{2}$$

where $A$ is the surface coverage of bryophytes and lichens in [$\mathrm{m}^2$ cover $\mathrm{m}^{-2}$ ground], $G$ is net growth in [$\mathrm{m}^2$ new cover $\mathrm{m}^{-2}$ cover], accumulated over a month, and $\eta_{\mathrm{E}}$ is a dimensionless "expansion efficiency" which is set to 0.85. This value is taken from the

global, stand-alone version of LiBry, which has also been updated to a dynamic disturbance scheme. It was selected to obtain a realistic global distribution of surface coverage. The minimum statement ensures that the cover increment cannot exceed the

available area. The term $A(1.0 - A)$ describes limitation by both existing cover and free area available for growth. Reduction of cover is calculated as:

$$R_{\mathrm{G}} = \begin{cases} 0 & \text{if } G \geq 0 \\ min\left(-GA, A\right) & \text{if } G < 0 \end{cases} \quad (3)$$

and

$$R_{\mathrm{D}} = DA \quad (4)$$

where $D$ is the fraction of cover lost due to disturbance. It is set to 0.00083 per month. This value corresponds to a fire return interval of 100 years, which is characteristic for the boreal forest (Bonan and Shugart, 1989; Beer et al., 2006; Mouillot and Field, 2005). The minimum statement in Eq. 3 ensures that the cover reduction cannot lead to negative cover.

Not only the scheme for computing surface coverage, but also the representation of the organisms' physiological variation had to be modified to integrate LiBry into JSBACH. In the original version of the model, a Monte-Carlo approach is used to sample broad ranges of possible parameter values. From that, artificial "species" are generated that represent the functional diversity of bryophytes and lichens. The "species" perform differently under a given climate. Those which cannot maintain a cover larger than zero cannot "survive" in the model and the remaining ones are used to compute the productivity. This "species"-based approach is conceptually different from the traditional plant functional type (PFT) approach used in many land surface models, and also in JSBACH. To adapt LiBry to the PFT design of JSBACH, the number of artificial "species" is set to one. This "species" is then parameterized to correspond to boreal, ground-based bryophytes and lichens.

In general, PFTs in JSBACH are not able to coexist in the same place since they are represented in distinct, non-overlapping tiles. The tiles specify which fraction of a model grid cell is occupied by a certain PFT, thereby representing sub-gridscale heterogeneity. The bryophyte and lichen PFT, however, should be able to grow in combination with various different PFTs. It is therefore not limited to a specific tile, but instead it is implemented as an additional layer on top of the soil. This layer is available on tiles which contain trees or grasses, while it is excluded from tiles covered by crops or glaciers. The surface coverage of the bryophyte and lichen PFT may differ between the tiles of a given grid cell. This can result from the differential influences of the tiles' vegetation types on bryophyte and lichen growth, for instance due to differing LAI between the tiles.

Defining the bryophyte and lichen PFT as a layer in JSBACH has implications for the representation of the organisms' hydrological properties. The original version of LiBry computes the water content of the organisms through the balance of water input and loss, where the specific storage capacity for water per biomass is sampled for each "species" from a range of possible values from the literature. These values are not related to the geometry of the bryophyte or lichen. In JSBACH, however, the water storage capacity of a layer is determined by multiplying layer thickness by the porosity of the layer. Hence, to be consistent with JSBACH, the water storage capacity of the bryophyte and lichen PFT is computed from the thickness and porosity of the bryophyte and lichen layer. Based on the study by Soudzilovskaia et al. (2013), we set thickness to 4.5 cm, which corresponds to the median of the measured values. The measurements are based on green and undecomposed brown

tissue of bryophyte mats. Porosity is not directly measured in Soudzilovskaia et al. (2013), but they provide values of maximum volumetric moisture. Since we are actually interested in the water storage capacity of the bryophyte and lichen layer, we set the "effective" porosity to 80 %, which is at the higher end of the measured values of volumetric moisture.

In JSBACH, two thermal properties of the bryophyte and lichen layer have to be known to derive its influence on soil temperature: Thermal conductivity and heat capacity. Both are strongly dependent on the relative moisture content of the bryophyte and lichen layer. Moreover, they depend on the state of matter of the water in the bryophytes and lichens, which can be liquid or frozen in various relative amounts. In analogy to the soil layers in JSBACH (Ekici et al., 2014), we write the dependence of the thermal conductivity of the bryophyte and lichen layer, $\kappa$, on water content as:

$$\kappa = \kappa_{\mathrm{o}}^{(1.0-(v_{\mathrm{w}}+v_{\mathrm{i}}))} \kappa_{\mathrm{w}}^{v_{\mathrm{w}}} \kappa_{\mathrm{i}}^{v_{\mathrm{i}}} \mathrm{K_e} + (1.0 - \mathrm{K_e})\kappa_{\mathrm{d}} \tag{5}$$

where $\kappa_{\mathrm{o}}$ is the thermal conductivity of organic matter which is set to 0.25 [$\mathrm{W\,K^{-1}\,m^{-1}}$] (Beringer et al., 2001), $\kappa_{\mathrm{w}}$ is the thermal conductivity of liquid water, $\kappa_{\mathrm{i}}$ is the thermal conductivity of ice and $\kappa_{\mathrm{d}}$ is the thermal conductivity of the dry bryophyte and lichen cover, which is set to 0.05 [$\mathrm{W\,K^{-1}\,m^{-1}}$] according to (O'Donnell et al., 2009). $v_{\mathrm{w}}$ and $v_{\mathrm{i}}$ are the volumetric moisture and ice contents of the bryophyte and lichen cover which are calculated by dividing the absolute water or ice content in [m] by he thickness of the cover. $\mathrm{K_e}$ is the Kersten number which is calculated for the linear regime as:

$$\mathrm{K_e} = min(1.0, \frac{v_{\mathrm{w}} + v_{\mathrm{i}}}{\epsilon}) \tag{6}$$

where $\epsilon$ is the porosity of the bryophyte and lichen cover. The dependence of the heat capacity of the bryophyte and lichen cover $C$, on water content is written as:

$$C = (1.0 - \epsilon)C_{\mathrm{o}} + v_{\mathrm{w}}C_{\mathrm{w}}\rho_{\mathrm{w}} + v_{\mathrm{w}}C_{\mathrm{i}}\rho_{\mathrm{i}} \tag{7}$$

where $C_{\mathrm{o}}$ is the heat capacity of organic matter which is set to 2.5E6 [$\mathrm{J\,m^{-3}\,K^{-1}}$] (Beringer et al., 2001), $C_{\mathrm{w}}$ is the heat capacity of liquid water, $C_{\mathrm{i}}$ is the heat capacity of ice and $\rho_{\mathrm{w}}$ and $\rho_{\mathrm{i}}$ are the densities of liquid water and ice.

Figure 2 shows the thermal conductivity as well as the heat capacity of the modelled bryophyte and lichen cover in relation to the water saturation of the cover and the state of matter of the water. Both the curves for thermal conductivity as well as heat capacity of the cover compare well with measured values from e.g. Soudzilovskaia et al. (2013); Jorgenson et al. (2010); O'Donnell et al. (2009). This suggests that our chosen average values for thickness and porosity result in realistic thermal properties of the bryophyte and lichen cover.

To simulate the influence of the bryophyte and lichen layer on soil temperature, the layer and its thermal properties are included in the vertical heat transfer scheme of JSBACH. The original heat transfer scheme for permafrost soil, which is described in detail in Ekici et al. (2014), already contains an organic layer which represents bryophyte and lichen ground cover. This layer, however, has constant thermal properties since it does not consider the water content of bryophytes and lichens. The

thermal conductivity of this layer is set to 0.25 [W K$^{-1}$ m$^{-1}$] and its heat capacity is set to 2.5E6 [J m$^{-3}$ K$^{-1}$]. These values are shown in Fig. 2 for comparison. The constant layer is replaced by the new layer, which explicitly simulates water and ice content of bryophytes and lichens and the associated dynamic thermal conductivity and heat capacity (Fig. 2).

The heat transfer scheme subdivides the soil column into several layers, with the bryophyte and lichen layer on top. Additionally, a set of snow layers is simulated above the other layers in case snow is present. Thermal conductivity and heat capacity of each layer are then used to determine the vertical temperature profile by solving the equation for heat conduction for all layers. Thereby, the scheme determines the temperature of a layer in a given grid cell by first calculating the temperature profile for all tiles of the grid cell separately and then averaging the temperatures in each layer weighted by the area fraction of the tiles. The lower boundary condition of the scheme is set by assuming zero heat flux at the bottom of the soil column. The upper boundary condition is surface temperature, which is calculated from the surface energy balance using radiative forcing and the ground heat flux from the previous time step. The scheme also considers the influence of phase change of water on the temperature of a layer, thereby allowing to compute freezing and melting of water in the bryophyte and lichen layer.

Another important difference between the old and the new version of the heat transfer scheme is dynamic surface coverage of bryophytes and lichens. While the surface coverage of the old organic layer is 100 % everywhere, the coverage of the new bryophyte and lichen layer varies between 0 and 100 % between the grid cells of JSBACH (Eq. 1). This means that each tile in a grid cell has a part where the bryophyte and lichen layer is present and another part where the layer is absent. The vertical heat transfer scheme, however, requires a constant number of layers for each tile. Hence, in the new version of the scheme, the temperature profile is calculated twice for each tile, one time for all layers including the bryophyte and lichen layer, and another time leaving out the bryophyte and lichen layer. Subsequently, the average temperature of each layer of the respective tile is obtained by weighting the two profiles with their associated surface coverage. However, the two profiles have a different number of layers, not only due to the bryophyte and lichen layer, but also because the number of snow layers may differ between the parts. Consequently, we fill up the "empty" layers on top by using the surface temperature for the averaging procedure, according to Figure 3.

We implemented two further changes to the heat transfer scheme described in Ekici et al. (2014): We increased the number of soil layers from 5 to 7, thereby extending the layer boundaries at 6.5, 32, 123, 413 and 983 cm depth by boundaries at 23.04 and 53.18 m depth. This ensures that biases due to the zero heat flux lower boundary conditions are negligible and temperature of the lowest layer is not fluctuating after spin-up. Moreover, we complement the snow scheme by dynamic formulations of snow density (Verseghy, 1991) and thermal conductivity (Goodrich, 1982), which take into account the effect of changes in snow density with time on the thermal conductivity of the snow layer. By considering the dynamic nature of snow thermal conductivity, we make the snow layers consistent with the other dynamic layers of the vertical heat transfer scheme in JSBACH. Note that compaction of the bryophyte and lichen layer by snow is not considered here, since the effect of this process on overall thermal conductivity are probably small.

## 2.2 Model setup

The advanced JSBACH model is forced by observation-based gridded climate fields at 0.5°pixel size and daily resolution during the period 1901-2010 following Beer et al. (2014). The grid cells have a size of approximately 30 x 30 km at 60° N and they are divided into four tiles according to the four most dominant plant functional types of this grid cell. This vegetation coverage is assumed to stay constant over the time of simulation. In the model simulations used in this study, we apply new soil parameters. Hydrological parameters have been assigned to each soil texture class following Hagemann and Stacke (2015) according to the percentage of sand, silt and clay at 1 km spatial resolution as indicated by the Harmonized World Soil Database (FAO et al., 2009). Thermal parameters have been estimated as in (Ekici et al., 2014) at the 1 km spatial resolution. Then, averages of 0.5°grid cells have been calculated. Soil depth until bedrock follows the map used in Carvalhais et al. (2014) based on Webb et al. (2000). The time step of JSBACH is set to 30 minutes. In our simulation, we only consider the region north of 50° N, since most of the world's permafrost soils are located in this region and it is thus possible to save a substantial amount of computation time.

We use a step-wise approach to run JSBACH into a steady-state, before we continue the simulation with a transient run leading to current climatic conditions. First, a 50-year hydrothermal spin-up is performed, where freezing and thawing of water is switched off, so that only liquid soil water occurs and develops into a steady-state with climate. This is done to prevent water in the deep soil layers from freezing before the water content had time to deviate from the initial conditions, since the initial values may not necessarily be realistic. Since we do not want to include the influences of a climate change in the spin-up, these first 50 years are run with WATCH climate data randomly selected from the time period 1901 to 1930. Atmospheric $CO_2$ is set to a preindustrial value of 285 ppmv. Subsequently, the spin-up run is continued with a 100-year simulation, which is used to generate a steady-state of soil ice, water and temperature. Therefore, freezing/thawing is switched on this time. Again, this run uses random climate data from 1901 to 1930 and preindustrial $CO_2$. The hydrothermal spin-up is followed by a 5000-year simulation of the CBALANCE model, which is a simplified version of JSBACH that simulates only the slow carbon pools of soil and vegetation as a function of the NPP of the vegetation. By using the NPP output of the last 30 years of the hydrothermal spin-up and repeating it for 5000 years in CBALANCE, soil carbon is run into a steady-state. After that, the simulation is continued with a 50-year run from 1851 to 1900, where the climate data are again randomly selected from the 1901 to 1930 period, but atmospheric $CO_2$ content is transient. Finally, the simulation is completed by a fully transient run from 1901 to 2010, where climate data are taken from Beer et al. (2014). The model is forced with atmospheric $CO_2$ concentrations during 1851-2010 following Meinshausen et al. (2011).

We perform spin-up and transient runs for 3 different configurations of the new JSBACH model version: a) the standard configuration which includes the process-based bryophyte and lichen layer, b) a configuration, where the bryophyte and lichen layer has 100 % surface coverage and constant thermal properties, corresponding to the organic layer in the old permafrost version of JSBACH and c) a configuration, where no bryophyte and lichen layer is simulated. These 3 simulations will be referred to as "Dynamic", "Constant" and "Without" throughout the text. By comparing the 3 simulations we can assess the impact of the dynamic coverage and thermal properties of the bryophyte and lichen layer on soil temperature.

To assess how uncertainty in the parameterization of the bryophyte and lichen PFT affects our estimated difference in soil temperature between the "Dynamic" and "Without" simulations, we run a sensitivity analysis. We test several bryophyte and lichen parameters which can affect productivity and, consequently, surface coverage and the associated thermal properties of the bryophyte and lichen layer. A detailed description of the sensitivity analysis can be found in Sect. A in the appendix.

To evaluate our modelling approach, we compare simulated bryophyte and lichen surface coverage and NPP averaged over the study region to field measurements. Since large-scale observations are not available to our knowledge, we estimate "characteristic" values of surface coverage and NPP for the study region based on small-scale measurements (see Sect. B in the appendix for details).

## 3 Results

### 3.1 Effects of bryophytes and lichens

Large-scale patterns of net primary productivity (NPP) and surface coverage of the bryophyte and lichen ground cover simulated in JSBACH are shown in Fig. 4. The maps are based on average values over the last 15 years of a transient "Dynamic" JSBACH simulation from 1901 to 2010, where dynamic thermal properties of the bryophyte and lichen layer are taken into account (see Sect. 2.2 for details).

The spatial pattern of NPP by bryophytes and lichens in JSBACH (Fig. 4 a) ) can be structured into areas with high productivity of over $100 \, \mathrm{g \, m^{-2} \, a^{-1}}$ of carbon, areas of intermediate productivity of 40 to $100 \, \mathrm{g \, m^{-2} \, a^{-1}}$ and areas of low productivity of less than $40 \, \mathrm{g \, m^{-2} \, a^{-1}}$. It should be noted, however, that also in areas of low productivity NPP usually exceeds $20 \, \mathrm{g \, m^{-2} \, a^{-1}}$. Highly productive areas are found in North Scandinavia, the north-european part of Russia, North-West Siberia, Kamchatka, parts of West Alaska and parts of East Canada. Areas of intermediate productivity extensively surround the highly productive ones in general. Less productive areas are found in the southern parts of Russia, in Middle and East Siberia and in parts of Central Canada. This spatial pattern may be explained by a combination of rainfall and available photosynthetically active radiation (PAR) on the ground (see Fig. 10 in the appendix): Regions such as North Scandinavia, North-West Russia, Kamchatka, Alaska and East Canada exhibit relatively high precipitation, but simulated above-ground vegetation is sparse enough to allow for moderate levels of light on the ground. This promotes high productivity of bryophytes and lichens. South Scandinavia, parts of east-european Russia or the region south of the Hudson Bay, however, show relatively high precipitation together with little availability of PAR on the ground, which limits productivity. Large regions in North Canada and East Siberia, in turn, exhibit a high PAR availability on the ground due to sparse vegetation, but precipitation is very low there, which again limits bryophyte and lichen growth.

The bryophyte and lichen layer is excluded from JSBACH-tiles containing cropland (Sect. 2.1). Consequently, productivity of bryophytes and lichens is strongly limited by available area in regions with extensive agriculture, such as South Russia and South Canada. This explains why NPP is low in these regions in spite of favorable climatic conditions for growth.

The spatial distribution of simulated bryophyte and lichen surface coverage (Fig. 4 b) ) correlates in general with the spatial pattern of NPP. This is due to the uniform disturbance interval of 100 years prescribed in our simulation for the whole study

region. With constant disturbance, variation in the growth of the bryophyte and lichen cover results only from variation in the organisms' NPP (see Eq. 1). However, some areas in Russia and Canada show a high surface coverage in spite of only intermediate NPP, whereas other areas show less coverage for even higher values of NPP. This is due to surface coverage being plotted on a grid scale basis. If a significant part of the grid cell is covered by lakes or glaciers, coverage cannot reach 100 % although NPP might be very high.

Figure 5 shows the effect of the bryophyte and lichen layer on soil temperature, averaged over the whole study region, but limited to the biomes "boreal forest" and "tundra" as defined by Olson et al. (2001) (see also Fig. 10 c) ). We compare three 1901 to 2010 transient simulations with differing properties: The bryophyte and lichen layer with dynamic surface coverage and dynamic thermal properties ("Dynamic"), an organic layer with constant, uniform coverage and thermal properties ("Constant") and a setup with no bryophyte and lichen layer ("Without", see Sect. 2.2 for details). The annual mean temperature of the uppermost soil layer is shown in Fig. 5 a) for the last 15 years of the simulations. The dynamic bryophyte and lichen layer leads to a considerable decrease in topsoil temperature of 2.7 K in the model (Tab. 1), compared to the simulation without bryophyte and lichen layer. The effect of the organic layer with constant coverage and thermal properties is much smaller, only around 0.5 K, compared to the "Without" simulation.

In Fig. 5 b), the annual amplitude of topsoil temperature is shown, averaged over the last 15 years of the simulations. The annual decreasing effect of the bryophyte and lichen layer on soil temperature (Fig. 5 a) ) results from a strong decrease in summer that overrules the slight increase in soil temperature in winter. Both the dynamic as well as the constant bryophyte and lichen layer show this pattern. The organic layer with constant properties, however, should actually have no overall effect on annual soil temperature, since it dampens the heat flux between atmosphere and soil uniformly throughout the year. Hence, to explain this outcome, the additional effect of the snow layer on vertical heat transfer in JSBACH needs to be considered: In winter, the snow layer limits heat transfer already to such an extent, that the further dampening effect of the organic layer is secondary. Interestingly, the dampening effect of the dynamic bryophyte and lichen layer is not uniform throughout the year. In summer, the dynamic layer is a more efficient insulator than the constant layer, which can be explained by the additional reduction in thermal conductivity due to low moisture content. In winter, however, the dynamic layer is less efficient in insulating the soil than the constant one, due to the occurrence of ice inside the layer, which strongly increases its thermal conductivity (see also Fig. 2).

Figures 5 c) and d) display the vertical soil temperature profiles for July and January for the 7 soil layers simulated in JSBACH. In summer, the insulating effect of the bryophyte and lichen layer leads to a strong decrease in soil temperature in the topsoil, which becomes less pronounced with increasing soil depth. In winter, insulation by the bryophyte and lichen layer results in warmer topsoil temperatures. However, below 1 m depth, the seasonal dampening effect is not visible anymore and the overall decreasing effect of the bryophyte and lichen layer on soil temperature prevails.

In Figs. 6 a) and b) annual average topsoil temperature and active layer thickness estimated by JSBACH are shown, averaged for the years 1996 to 2010 of the "Dynamic" simulation. Soil temperature decreases with increasing latitude and the lowest values outside of Greenland are found in North-East Siberia, particularly in the East-Siberian mountains. The spatial pattern of active layer thickness is similar to soil temperature. Large parts of North-East Siberia and North Canada exhibit shallow active

layer thickness of less than 1 m while the regions adjacent to the South show a deeper active layer of up to 3 m. By definition, we did not include regions with more then 3 m of active layer thickness in our analysis of permafrost soil. These regions are coloured in white in Fig. 6 b).

Figures 6 c) and d) show the large-scale pattern of differences in topsoil temperature and active layer thickness between the "Dynamic" and the "Without" simulation. The general pattern of soil temperature difference is similar to surface coverage, which makes sense since it is the bryophyte and lichen ground cover which reduces soil temperature. However, in mountainous regions, the reducing effect of the moss and lichen cover on soil temperature seems to be stronger than in flat terrain, indicating a non-linear response of the insulating effect to the climate forcing or to soil properties. The spatial differences in soil temperature span a range of 0 to 5.7° C.

The reduction in active layer thickness resulting from the insulating effect of the bryophyte and lichen layer is substantial, particularly at the center of the permafrost area, in Siberia and North Canada. However, also in the southern regions the reduction in soil temperature is large enough to decrease active layer thickness below the 3 m-threshold for permafrost soil. For this reason, the areal extent of permafrost soil in Fig. 6 b) is considerably larger than the coloured area in Fig. 6 d). Estimated permafrost area as defined by an active layer thickness of less than 3 m increases due to the insulating effect of the bryophyte and lichen layer by 32 % from 14.8 million km$^2$ to 19.5 million km$^2$. This new result is much more comparable to the 22 million km$^2$ permafrost area reported for the Northern Hemisphere (French, 2007, p. 95). On average, active layer thickness is reduced by 107 cm for the study region with a spatial range of 0 to 224 cm.

Average values over the study region are listed in Tab. 1. They show a significant impact of bryophytes and lichens on soil temperature and active layer thickness with a large spatial variation.

The outcomes of the sensitivity analysis are shown in Tab. 2 in Sect. A in the appendix. The general result is that the impact of the simulated bryophyte and lichen layer on the temperature difference between the "Dynamic" and "Without" simulations is robust under a large range of parameter values of the bryophyte and lichen PFT. Moreover, if several parameters are varied at the same time, their combined effect on the soil temperature difference is less than the sum of all individual effects.

The results of the model evaluation are shown in Sect. B in the appendix. Field measurements of NPP (Tab. 3) show a large variation, but the characteristic value of NPP for the study region is likely higher than 10 and lower than 100 g m$^{-2}$ a$^{-1}$ of carbon. Hence, our JSBACH estimate of 49 g m$^{-2}$ a$^{-1}$ of carbon (Tab. 1) compares reasonably well with the values from the field. Observations of surface coverage of bryophytes and lichens for the study region also show some variation (Tab. 4), but most likely the characteristic value lies in the range of 0.6 to 0.8. This suggests that our estimated average surface coverage of 0.69 is realistic. Comparison of simulated subsoil temperature and active layer thickness to observation-based maps for the region of Yakutia shows that JSBACH still underestimates subsoil temperature, same as previous model versions. This finding is expected since the bryophyte and lichen layer in JSBACH reduces soil temperature and consequently cannot reduce the cold bias. The estimate of active layer thickness, however, is improved in general since this variable was overestimated in the previous model version. The corresponding maps for Yakutia are shown in Fig. 9.

# 4    Discussion

In this study we quantified the reducing effect of the bryophyte and lichen ground cover on soil temperature at high latitudes at the large scale. For this purpose, we estimated dynamic surface coverage and thermal properties of bryophytes and lichens with a process-based model and we integrated this model into the global land surface model JSBACH.

We estimated an average decrease in temperature of the uppermost soil layer of 2.7 K with a spatial range of 0 to 5.7 K. This is a substantial effect, it has a similar size as the projected increase in global near-surface air temperature under the RCP 4.5 warming scenario of the Intergovernmental Panel on Climate Change (Pachauri et al., 2014). Our finding is consistent with various field experiments as well as modelling studies at site level, which confirm the important role of bryophytes and lichens for reducing heat exchange between atmosphere and soil at high latitudes (Beringer et al., 2001; Gornall et al., 2007; Jorgenson et al., 2010). Our results suggest that the insulating effect of the bryophyte and lichen ground cover should be taken into account in large-scale modelling studies which focus on feedbacks between permafrost soil and atmospheric $CO_2$ under climatic change.

Moreover, we showed that representing the dynamics of both surface coverage and thermal properties of bryophytes and lichens is crucial for estimating their insulating effect. Using a simple organic layer with constant coverage and neglecting the influence of water or ice content on thermal conductivity and heat capacity of the layer likely results in underestimating the decrease in soil temperature.

Alternatively to introducing a bryophyte and lichen layer with dynamic thermal properties, one might ask if calibrating a constant organic layer would also be sufficient to simulate the insulating effect. In Fig. 7 we show that there are qualitative differences between varying thermal conductivity of a constant organic layer and the dynamic bryophyte and lichen layer: If constant thermal conductivity is calibrated to reproduce soil temperature under a bryophyte and lichen layer in summer, the temperature in winter and, consequently, annual average temperature is too warm. To simulate an average annual soil temperature similar to that under the bryophyte and lichen layer, the constant thermal conductivity has to be strongly reduced, since the warming effect in winter has to be compensated. This leads to an unrealistically weak seasonal amplitude of soil temperature.

We want to point out that the bryophyte and lichen layer should not be confused with an organic soil layer from a modelling perspective. So far, JSBACH does not include an organic soil layer. Differences between these two types of layers are the relatively low thickness of the bryophyte and lichen layer compared to the organic layer, which can be several tens of centimeters thick due to the accumulation of peat. This means that the organic layer may have a significantly higher heat capacity when saturated with water or ice. Therefore, the reducing effect of high thermal conductivity on insulation can be counteracted by the large amount of energy required for a temperature change of the layer, which increases insulation. Furthermore, the higher residual water content of the peat layer compared to the bryophyte and lichen layer may result in a lower variability of thermal conductivity. These two differences may explain why e.g. Atchley et al. (2016) find a relatively small influence of water content on the insulating effect at the surface, since their model approach includes an organic layer.

Although the focus of our study is on soil temperature, the process-based bryophyte and lichen scheme in JSBACH also provides an estimate of the organisms' net primary productivity (NPP). The average simulated NPP of $49\,\mathrm{g\,m^{-2}\,a^{-1}}$ of carbon for the biomes "boreal forest" and "tundra" corresponds to roughly $10\,\%$ of average boreal forest NPP (Gower et al., 2001). This is a lower fraction than stated in the study of Turetsky et al. (2010), which estimate a contribution of $14\,\%$ to $58\,\%$ of moss to total ecosystem NPP for various boreal forest sites. The study by Goulden and Crill (1997), however, estimates a lower contribution of around $10\,\%$ by mosses to black spruce forest NPP. The variation in the fraction of forest NPP attributed to bryophytes and lichens can be explained by differences in hydrological conditions and vegetation structure between the sites. High values of productivity are mainly found on wetland sites, where high water and light availability sustain a productive *Sphagnum*-cover. JSBACH, however, is mainly designed to simulate upland soils of the boreal forest. Wetlands are not yet included and, consequently, the high productivity of mosses in these areas is not reflected in our estimate. Implementing a scheme for wetland hydrology and the associated additional supply of water for mosses from below would be a useful extension of JSBACH in the future.

The size of the insulating effect depended on the relations between the thermal properties of the bryophyte and lichen cover and its water or ice content. These relations are well established by field measurements and theory (see Fig. 2). Furthermore, our estimate of soil temperature was sensitive to the surface coverage of bryophytes and lichens, which, in our model, largely depends on the simulated NPP of the organisms. To assess how well JSBACH is able to represent current bryophyte and lichen NPP and surface coverage at high latitudes, we compared our estimates to field measurements. However, due to large variation in the field studies and a lack in up-scaled estimates, it was difficult to define characteristic values of NPP and surface coverage for the study region (see Tab. 3 and 4). Given the considerable uncertainty in the measurements, our estimates of bryophyte and lichen NPP and surface coverage agreed well with field observations. The enhancement of JSBACH due to representation of near-surface vegetation state and functions will improve the reliability of future projections of northern ecosystem responses to environmental change as well as climate-carbon cycle feedbacks in future studies.

We compared modelled subsoil temperature and active layer thickness to observations for the region of Yakutia. Our estimated active layer thickness matches reasonably well to observations and it is improved over the previous version of JSBACH (Ekici et al., 2014). This suggests that the dynamic bryophyte and lichen layer leads to a more realistic representation of vertical heat transfer in the model. JSBACH still underestimates subsoil temperature compared to the previous model version (Ekici et al., 2014). The cold bias is most pronounced for the East-Siberian mountains where bryophyte and lichen cover is relatively low (Fig. 4 b) ). Hence, it seems likely that the reason for the cold bias is not directly related to the bryophyte and lichen layer. A probable explanation for the cold bias is the relatively strong sensitivity of annual average soil temperature to the parameterization of the snow layer scheme in JSBACH. Improvements in the representation of snow may improve the simulated soil temperature.

Our modelling approach is designed to capture the most important properties of the bryophyte and lichen ground cover which influence heat exchange between atmosphere and soil. These properties include the dependency of bryophyte and lichen heat conduction on moisture and ice content as well as the relation of productivity and, consequently, surface coverage to various environmental factors. Moreover, indirect effects of the bryophyte and lichen cover on heat exchange are represented,

such as cooling of the surface due to evaporation from the organisms' thalli. However, there are several other relations between the bryophyte and lichen cover and the environment that may affect heat exchange at the surface, which we did not consider in this study for simplicity. It has been shown by Bernier et al. (2011), for instance, that conversion of black spruce forests into lichen woodlands in Canada results in atmospheric cooling due to the higher albedo of lichens compared to forest. Effects of

lichens and bryophytes on albedo and, consequently, on surface temperature have also been shown by Stoy et al. (2012), who discuss species-specific differences in these effects. In JSBACH, we did not include the effect of bryophytes and lichens on surface albedo, since the organisms vary considerably in their colour. Instead of assigning an arbitrary value for albedo to the bryophyte and lichen PFT in JSBACH, we thus assume that the albedo of the PFT is similar to the soil albedo. The evaporative cooling effect of the bryophyte and lichen cover on surface temperature may be modulated by water uptake from Larch trees,

which are able to root into the near-surface vegetation layer.

The bryophyte and lichen layer is represented by one single PFT in JSBACH, for reasons of consistency with vascular vegetation in JSBACH and also for computational efficiency. However, this lack of diversity may have consequences for the estimated effect of the bryophyte and lichen layer on soil temperature. Given similar climatic conditions, bryophyte and lichen species may differ in their degree of water saturation and, consequently, in their thermal properties. Lichens, for instance, have

in many cases a lower porosity and water storage capacity than bryophytes, particularly mosses (Schulze and Caldwell, 2012). This means that lichens may be less sensitive to water content than bryophytes regarding their thermal properties. However, both groups include species which do not match this pattern. If large regions differ in their dominant bryophyte or lichen species, this may affect our estimated patterns of soil temperature and active layer thickness. Additionally, bryophyte and lichen species differ in their thickness, while thickness within a species is relatively constant (Soudzilovskaia et al., 2013).

Since the bryophyte and lichen PFT in JSBACH has a constant thickness, we may underestimate further spatial effects on the soil thermal regime.

Another simplifying assumption in JSBACH is the uniform disturbance interval of 100 years for the whole study region. This value represents the average fire return interval in the boreal forest, where fire is the dominant process for disturbance (Mouillot and Field, 2005). The occurrence of fire, however, may vary between regions. Fires may be much less frequent in

parts of North Canada and North-East Siberia, for instance (Bonan and Shugart, 1989), which could lead to larger spatial differences in simulated bryophyte and lichen surface coverage. Under climatic change, shifts in vegetation distribution and increased temperatures may lead to changes in the fire interval at high latitudes. Consequently, including a more dynamic representation of disturbance in our approach would be particularly beneficial for modelling future bryophyte and lichen cover under scenarios of climate change.

**5   Conclusions**

Here we present an new version of the global land surface model JSBACH that estimates NPP, surface coverage and dynamic thermal properties of bryophytes and lichens through a process-based scheme. We apply JSBACH to quantify the impact of the bryophyte and lichen ground cover on the soil thermal regime at high latitudes. Thereby, we estimate a considerable average

cooling effect of the bryophyte and lichen cover of 2.7 K (0 to 5.7 K) for the uppermost soil layer. Furthermore, we find that the strength of the cooling effect largely depends on an accurate representation of dynamic coverage and thermal properties of bryophytes and lichens.

These results suggest that the reducing effect of the bryophyte and lichen ground cover on soil temperature should be accounted for in studies which aim at quantifying feedbacks between permafrost soil temperature and atmospheric $CO_2$ due to climate change. Since our process-based approach also allows for predicting future bryophyte and lichen surface coverage at high-latitude ecosystems, a potential next step is to simulate the future impact of bryophytes and lichens on active layer thickness and permafrost extent under a transient scenario of climate change.

## Appendix A: Sensitivity analysis

We perform a sensitivity analysis to assess how uncertainty in the parameter values chosen for the bryophyte and lichen PFT affects our estimates. In the following we describe several parameters which are characteristic for the bryophyte and lichen PFT and which may affect NPP, surface coverage or thermal properties of the PFT:

- The parameter porosity, $\epsilon$, of the bryophyte and lichen surface layer is set to $80\%$ in the model. This value is varied by around $20\%$ in each direction, from a minimum value of $65\%$ to a maximum of $95\%$. Porosity shows large natural variation in bryophytes and lichens, from $10\%$ in some lichen species (Valladares et al., 1993) to $99\%$ in some moss species (O'Donnell et al., 2009). We are interested, however, in uncertainty concerning the *average* porosity of the bryophyte and lichen ground cover for large regions. Hence, we do not vary $\epsilon$ for the full range since it is unlikely that the average porosity is close to an extreme value.

- The thickness $z$ of the simulated bryophyte and lichen layer is set to $4.5\,\mathrm{cm}$ in the model. It is defined as the undecomposed living and dead parts of a bryophyte or lichen mat. Observed values of thallus thickness show a large variation between species, ranging from less than a millimeter to several tens of centimeters (Nash III, 1996; Bell and Hemsley, 2011). Here, we vary $z$ from 1 to $10\,\mathrm{cm}$, which is a slightly larger range than determined by Soudzilovskaia et al. (2013) for 18 bryophyte species. We do not test extreme values of $z$, since we are interested in bryophyte and lichen species that are actually able to form a macroscopic ground cover. We do not, for instance, consider flat lichen crusts on boulder surfaces, since they are not likely to play a significant role for large-scale heat exchange between soil and atmosphere.

- The diffusivity of the water-saturated bryophyte or lichen thallus for $CO_2$, $D_{CO_2,sat}$, affects $CO_2$-uptake and, consequently, NPP and surface coverage of bryophytes and lichens in the model. We set $D_{CO_2,sat}$ to a value of 0.01 $[\mathrm{mol\,m^{-2}\,s^{-1}}]$ based on (Williams and Flanagan, 1998) and vary this value by multiplying it by the factors 0.5 and 2.0. We choose this form of variation since $D_{CO_2,sat}$ shows relatively large natural variation from around 5E-4 to 2E-2 $[\mathrm{mol\,m^{-2}\,s^{-1}}]$ (Williams and Flanagan, 1998; Cowan et al., 1992) and, consequently, a linear variation would not be adequate (Porada et al., 2013). It should be noted that variation in $D_{CO_2,sat}$ represents an extension to the original bryophyte and lichen model, described in Porada et al. (2013).

- The model parameter critical water saturation, $\Theta_{crit}$, determines, which value of saturation is necessary for the bryophyte and lichen layer to reach full metabolic activity. The increase from zero saturation and no activity to $\Theta_{crit}$ is assumed to be linear (Porada et al., 2013). Here, $\Theta_{crit}$ is set to $30\%$, which corresponds to the lower end of the range of possible values (Porada et al., 2013) and reflects the relatively fast activation of common boreal forest floor mosses (Williams and Flanagan, 1998). Hence, we vary $\Theta_{crit}$ by setting it closer to the upper bound, to a value of $60\%$.

- The specific maintenance respiration rate $R_{main}$ varies over 3 orders of magnitude between different bryophyte and lichen species (Porada et al., 2013). For LiBry in JSBACH, we chose an intermediate value of 1.5E-6 $[\mathrm{mol\,m^{-2}\,s^{-1}}]$ and due to the large range of possible values we vary $R_{main}$ by multiplication by the factors 0.5 and 2.0. In the model, $R_{main}$

is related to photosynthetic capacity and biomass turnover rate of the bryophyte and lichen layer via the parameter "Ratio of Rubisco content to maintenance respiration", $\Phi_{\mathrm{RR}}$, which describes the tradeoff between photosynthetic capacity and respiration (Porada et al., 2013). LiBry has been shown to be sensitive to this parameter, so we vary the standard value of 5 [s] by 20 % in each direction.

- In addition to photosynthetic capacity, LiBry contains 4 parameters to calculate photosynthesis, which vary between different species (Porada et al., 2013): The enzyme activation energies $E_{\mathrm{a,Kc}}$ and $E_{\mathrm{a,Ko}}$, which control the temperature response of the Michaelis-Menten constants of the carboxylation and oxygenation reactions of Rubisco, and the molar carboxylation and oxygenation rates of Rubisco, $V_{\mathrm{M,C}}$ and $V_{\mathrm{M,O}}$. For LiBry in JSBACH, $E_{\mathrm{a,Kc}}$ is set to 120000 [J mol$^{-1}$], which is close to the upper bound of observed values, and $E_{\mathrm{a,Ko}}$ is set to 10000 [J mol$^{-1}$], which is a relatively low value regarding the observations (Porada et al., 2013). Hence, we vary $E_{\mathrm{a,Kc}}$ by setting it to the lower bound of 30000 [J mol$^{-1}$], and $E_{\mathrm{a,Ko}}$ by setting it close to the upper bound of 55000 [J mol$^{-1}$]. The molar carboxylation rate of Rubisco, $V_{\mathrm{M,C}}$, varies by two orders of magnitude (Porada et al., 2013). We therefore vary the intermediate value of 1.5 [s$^{-1}$] used here by multiplying it by the factors 0.5 and 2.0. The molar oxygenation rate of Rubisco, $V_{\mathrm{M,O}}$, is set here to 2.2 [s$^{-1}$], close to the observed maximum value. We vary it by setting it to 0.5 [s$^{-1}$], close to the observed minimum.

- Photosynthesis and respiration in LiBry are related to surface temperature via the two parameters optimum temperature of photosynthesis, $T_{\mathrm{opt}}$, and the Q$_{10}$-value of respiration, $Q_{10}$. Here, $T_{\mathrm{opt}}$ is set to 22° C and it is varied in both directions by 5° C. $Q_{10}$ is set to 2.4 and it is varied by 0.3 in both directions.

- Furthermore, we test how sensitive the effect of the bryophyte and lichen layer on soil temperature reacts to the disturbance interval $\tau_{\mathrm{D}}$ set in the model. We therefore vary the standard value of 100 years for $\tau_{\mathrm{D}}$ by multiplying it with the factors 0.5 and 2.0.

- Finally, we vary the expansion efficiency $\eta_{\mathrm{E}}$, which is set to 0.85, by 20 % in each direction.

For each varied parameter we run a step-wise simulation consisting of a steady-state spin-up and a transient run, as described above. For reasons of computational speed we run the model only on a single grid cell. The location of the grid cell is 55° 30' N, 98° 30' W, which roughly corresponds to the northern study area of the BOREAS project (e.g. Gower et al. (1997)). We then compare the difference in soil temperature of the uppermost soil level between the standard simulation and each of the simulations with varied parameters to quantify the impact of parameter uncertainty on our overall estimates. Moreover, we run additional simulations with more than one varied parameter to assess their combined effect on our estimates.

Many of the tested parameters have no strong effect on NPP and surface coverage of the simulated bryophyte and lichen layer. Thus, they do not affect the difference in topsoil temperature resulting from the dynamic layer. The following parameters, however, lead to changes in the temperature difference:

- The porosity $\epsilon$ of the bryophyte and lichen layer affects the temperature difference in both directions. Interestingly, a reduction in porosity leads to a reduced coverage in spite of slightly increased NPP. The reason for this is that lower

porosity results in a smaller specific area of the bryophyte and lichen PFT. Since net growth of cover (Eq. 1) is calculated by multiplying NPP with specific area, the effect of reduced specific area may overrule an increase in NPP. This leads to less cover expansion. The inverse effect occurs for higher porosity.

– The thickness $z$ affects the temperature difference, to a moderate extent. The slight increase in coverage for a lower value of $z$ mainly results from increased specific area. Comparing the parameters in Tab. 2, the slightly higher coverage cannot explain by itself the increase in temperature difference. It is probably the decrease in heat capacity associated with a thinner bryophyte and lichen layer which leads to a warmer and, consequently, drier layer in summer which acts as a more efficient insulator. The opposite then happens for a thicker layer. This means that the reducing effect of low moisture content on thermal conductivity of the bryophyte and lichen layer overrules the increasing effect of higher thickness in the model.

– Reducing the expansion efficiency $\eta_E$ also reduces surface coverage and, consequently, the soil temperature difference. However, the model is not very sensitive to this parameter, $\eta_E$ would have to be quite low to significantly affect our estimates.

– Increasing critical water saturation $\Theta_{crit}$ has a relatively strong decreasing effect on the temperature difference, since the associated slower activation of bryophyte and lichen photosynthesis significantly reduces NPP and surface coverage.

– The specific maintenance respiration rate $R_{main}$ has the strongest influence on the temperature difference compared to the other parameters, but only in one direction. A doubling of $R_{main}$ significantly reduces the temperature difference, whereas halving $R_{main}$ does not have any effect. The reason for this is that our standard value of $R_{main}$ is close to optimal, meaning that NPP decreases in both directions. The associated biomass turnover, however, increases linearly with $R_{main}$ and it has a strong reducing impact on net growth and cover expansion. This means that low turnover compensates for low NPP at low $R_{main}$, but high turnover has an additional reducing impact at high $R_{main}$. The "Ratio of Rubisco content to maintenance respiration", $\Phi_{RR}$, has only a minor impact on the temperature difference.

– Halving the molar carboxylation rate of Rubisco, $V_{M,C}$, reduces significantly the temperature difference due to strongly decreased NPP and coverage. Doubling $V_{M,C}$, however, does not have any effect on temperature due to the nonlinear response of coverage on increased NPP. The activation energy $E_{a,Kc}$ has a moderate reducing impact on NPP, coverage and thus on the temperature difference.

– Reducing the optimum temperature of photosynthesis, $T_{opt}$, by $5°$ C results in significantly lower NPP, coverage and temperature difference, while increasing $T_{opt}$ has no significant effect. The reason for the asymmetric response of NPP to $T_{opt}$ is the exponential dependence of respiration on the, in this case, mostly negative difference between surface temperature and $T_{opt}$. Since $T_{opt}$ is already relatively high, a further increase leads only to a small reduction in respiration. Lowering $T_{opt}$, however, results in a strong increase in respiration and, consequently, reduced NPP.

To assess the effect of combined varied parameters, we select four parameters which significantly affect the estimated difference in topsoil temperature between the "Dynamic" and "Without" simulations (see Tab. 2). The outcome of this analysis is shown in Fig. 8. It can be seen that the combined effect of parameters is not additive, but becomes weaker with each additional parameter.

## 5  Appendix B: Model evaluation

To evaluate our modelling approach, we compare the JSBACH estimates of bryophyte and lichen surface coverage and NPP to field observations. However, to our knowledge, there are no field measurements of bryophyte and lichen surface coverage and NPP which cover the global scale. Therefore, we compile a list of available small-scale observations from various studies, which are largely taken from overviews in Turetsky et al. (2010) and Bona et al. (2013). We do not attempt to create a comprehensive review of measurements of surface coverage and NPP in high-latitude regions. However, our list is sufficient to establish "characteristic" values of surface coverage and NPP for the region north of 50° N. Thereby, we constrain our analysis to the biomes "boreal forest" and "tundra" (after Olson et al. (2001), see also Fig. 10 c) ). Since JSBACH is not primarily designed to simulate peatlands, we exclude studies which measure surface coverage and in peatlands. Several studies contain measurements both on peatland and upland sites. In this case, we do not consider values from peatlands. In many cases, the original studies provide NPP measurements in grams of biomass. We convert these values to grams of carbon using a factor of 0.45 (Bauer et al., 2009).

We compare the characteristic values of measured NPP to the average bryophyte and lichen surface coverage and NPP simulated by JSBACH, also constrained to boreal forest and tundra. Furthermore, we compare simulated soil temperature and active layer thickness to large-scale, observation-based maps for the region of Yakutia.

Table 3 shows studies which provide measurements of bryophyte and lichen NPP for the biomes "boreal forest" and "tundra". To obtain a characteristic value for NPP, we calculate the median NPP of each study and from this set of values we take again the median, which is then shown in the last row of Tab. 3 together with the range of the median values from all studies. Given the large variation in these median values of two orders of magnitude, it is difficult to constrain NPP in the study region based on field measurements. The average value probably lies somewhere between 10 and 100 $\mathrm{g\,m^{-2}\,a^{-1}}$ of carbon.

In Tab. 4 several studies are listed which measure surface coverage of bryophytes and lichens for boreal forest and tundra. Same as with NPP, we show in the last row the range and median of all studies' median values to obtain a characteristic surface coverage for the region. The range of observed surface coverage is large, but half of the (rounded) values lie between 0.6 and 0.8. We want to point out that the study by Rapalee et al. (2001) provides a large-scale estimate of surface coverage, based on remote sensing. They cover the whole BOREAS region which has an area of approximately 500000 $\mathrm{km^2}$ and arrive at a moss surface coverage of 0.57. This value agrees well with our estimate, given that the BOREAS region has a slightly lower than average simulated bryophyte and lichen coverage (Fig. 4).

Figure 9 compares the spatial patterns of subsoil temperature and active layer thickness estimated by JSBACH to observations from Beer et al. (2013) for the region of Yakutia. JSBACH underestimates subsoil temperature to a similar extent as the

version presented in Ekici et al. (2014). Potential reasons for the mismatch are discussed in Ekici et al. (2014) and include a reduced spatial heterogeneity of the observational map or biases in the representation of climate, soil properties or snow depth in JSBACH. The estimate of active layer thickness, however, is improved in general compared to the previous version of JSBACH, with the exception of the East-Siberian mountain range. The underestimation of active layer thickness there likely follows from the strong underestimation of soil temperature in this area. The overestimation of active layer thickness in the southern part of Yakutia may result from the patchy structure of the permafrost area in these regions, as discussed by Ekici et al. (2014).

**Appendix C: Additional output**

*Author contributions.* P. Porada developed the model code and prepared the manuscript with contributions from all co-authors, P. Porada and C. Beer designed the simulation setup and performed the simulations.

*Acknowledgements.* This work has been supported by the PAGE21 project, grant agreement number 282700, funded by the EC seventh Framework Programme theme FP7-ENV-2011, and the CARBOPERM project, grant agreement number 03G0836B, funded by the BMBF (German Ministry for Science and Education). The authors thank the Bolin Centre for Climate Research for financial support. Stockholm University and the National Supercomputer Centre at Linköping University provided computational resources. The authors thank three anonymous referees for their thorough and helpful reviews.

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

**Table 1.** Net primary productivity (NPP) and surface coverage of the simulated dynamic bryophyte and lichen layer, as well as differences between the "Dynamic" and "Without" simulations with regard to topsoil temperature and active layer thickness. The values are based on the last 15 years of 1901 to 2010 transient simulations and the region considered is limited to boreal forest and tundra. "g C" stands for "gram carbon".

| Variable | Average | Range | Unit |
| --- | --- | --- | --- |
| Bryophyte & lichen NPP | 49 | 0 – 192 | $\mathrm{g\,C\,m^{-2}\,a^{-1}}$ |
| Bryophyte & lichen surface coverage | 0.69 | 0 – 1.0 | $\mathrm{m^2\,m^{-2}}$ |
| Difference in soil temperature | -2.7 | 0 – -5.7 | K |
| Difference in active layer thickness | -107 | 0 – -224 | cm |

**Table 2.** Sensitivity of the estimated topsoil temperature difference due to the addition of a dynamic bryophyte and lichen layer to the parameterization of the bryophyte and lichen PFT. See Sect. 2.2 for a description of the parameters. $\Delta T$ is the temperature difference in the uppermost soil layer between the "Dynamic" simulation and the "Without" one (Sect. 2.2). NPP is in $\mathrm{g\,m^{-2}\,a^{-1}}$ of carbon. $\Delta P$ stands for a change in a model parameter, either by multiplication with a factor (* symbol), percentage change or by setting the parameter to the upper boundary (u.b.) or lower boundary (l.b.). $\Delta T/\Delta P$ is the percentage change in temperature difference ($\Delta T$) due to a change in a model parameter ($\Delta P$), compared to the control run. The sensitivity analysis is performed on a single grid cell of JSBACH (see Sect. 2.2 for details).

| Parameter | Value | $\Delta T$ | Coverage | NPP | $\Delta P$ | $\Delta T/\Delta P$ |
|---|---|---|---|---|---|---|
| $D_{\mathrm{CO_2,sat}}$ | 0.005 $[\mathrm{mol\,m^{-2}\,s^{-1}}]$ | -2.7 | 0.81 | 30.8 | *0.5 | 0 % |
| $D_{\mathrm{CO_2,sat}}$ | 0.02 $[\mathrm{mol\,m^{-2}\,s^{-1}}]$ | -2.7 | 0.82 | 32.7 | *2 | 0 % |
| $\tau_{\mathrm{D}}$ | 200 [a] | -2.7 | 0.82 | 32.1 | *0.5 | 0 % |
| $\tau_{\mathrm{D}}$ | 50 [a] | -2.7 | 0.82 | 32.1 | *2 | 0 % |
| $\eta_{\mathrm{E}}$ | 0.7 | -2.6 | 0.78 | 32.1 | -20 % | -4 % |
| $\eta_{\mathrm{E}}$ | 1.0 | -2.7 | 0.84 | 32.2 | +20 % | 0 % |
| $\epsilon$ | 65 % | -2.0 | 0.68 | 32.7 | -20 % | -26 % |
| $\epsilon$ | 95 % | -2.9 | 0.86 | 30.2 | +20 % | +7 % |
| $E_{\mathrm{a,Kc}}$ | 30000 $[\mathrm{J\,mol^{-1}}]$ | -2.4 | 0.75 | 25.6 | l.b. | -11 % |
| $E_{\mathrm{a,Ko}}$ | 50000 $[\mathrm{J\,mol^{-1}}]$ | -2.7 | 0.81 | 30.2 | u.b. | 0 % |
| $R_{\mathrm{main}}$ | 7.5E-7 $[\mathrm{mol\,m^{-2}\,s^{-1}}]$ | -2.7 | 0.80 | 25.6 | *0.5 | 0 % |
| $R_{\mathrm{main}}$ | 3.0E-6 $[\mathrm{mol\,m^{-2}\,s^{-1}}]$ | -1.6 | 0.53 | 25.9 | *2 | -41 % |
| $Q_{10}$ | 2.1 | -2.7 | 0.80 | 29.5 | -12.5 % | 0 % |
| $Q_{10}$ | 2.7 | -2.7 | 0.83 | 34.1 | +12.5 % | 0 % |
| $\Phi_{\mathrm{RR}}$ | 4.0 | -2.6 | 0.80 | 26.3 | -20 % | -4 % |
| $\Phi_{\mathrm{RR}}$ | 6.0 | -2.7 | 0.83 | 36.7 | +20 % | 0 % |
| $\Theta_{\mathrm{crit}}$ | 60 % | -2.0 | 0.67 | 16.9 | u.b. | -26 % |
| $T_{\mathrm{opt}}$ | 17 $[^\circ\mathrm{C}]$ | -2.2 | 0.70 | 24.3 | -5K | -19 % |
| $T_{\mathrm{opt}}$ | 27 $[^\circ\mathrm{C}]$ | -2.7 | 0.83 | 32.4 | +5K | 0 % |
| $V_{\mathrm{M,C}}$ | 0.75 $[\mathrm{s^{-1}}]$ | -2.0 | 0.66 | 14.1 | *0.5 | -26 % |
| $V_{\mathrm{M,C}}$ | 3.0 $[\mathrm{s^{-1}}]$ | -2.7 | 0.84 | 45.1 | *2 | 0 % |
| $V_{\mathrm{M,O}}$ | 0.5 $[\mathrm{s^{-1}}]$ | -2.7 | 0.82 | 32.1 | l.b. | 0 % |
| $z$ | 10 mm | -3.0 | 0.83 | 36.8 | l.b. | +11 % |
| $z$ | 100 mm | -2.4 | 0.80 | 28.8 | u.b. | -11 % |
| Control run | - | -2.7 | 0.82 | 32.1 | - | - |

**Table 3.** Studies which measure NPP in the biomes "boreal forest" and "tundra". Median and range of all values listed in a study are shown, "#" stands for the number of values. NPP is in $\mathrm{g\,m^{-2}\,a^{-1}}$ of carbon. "BOREAS N or S" stands for the northern and southern site of the BOREAS project (e.g. Gower et al. (1997)). BOREAS N is located in Manitoba, Canada and BOREAS S is located in Saskatchewan, Canada. The last row shows median and range of the values in the first column.

| Median | Range | # | Study | Location |
|---|---|---|---|---|
| 21 | 20 - 26 | 4 | (Camill et al., 2001) | BOREAS N |
| 35 | 0 - 144 | 7 | (Bond-Lamberty et al., 2004) | BOREAS N |
| 11 | 0 - 16 | 7 | (Mack et al., 2008) | Central Alaska |
| 42 | 40 - 44 | 2 | (Oechel and Van Cleve, 1986) | Central Alaska |
| 30 | 29 - 31 | 2 | (Ruess et al., 2003) | Central Alaska |
| 14 | 13 - 17 | 4 | (Vogel et al., 2008) | Alaska/Saskatchewan |
| 22 | 12 - 60 | 3 | (Schuur et al., 2007) | Central Alaska |
| 30 | 2 - 68 | 4 | (Shaver and Chapin III, 1991) | North Alaska |
| 25 | - | 1 | (Bisbee et al., 2001) | BOREAS S |
| 12 | - | 1 | (Gower et al., 1997) | BOREAS N/S |
| 104 | - | 1 | (Swanson and Flanagan, 2001) | BOREAS S |
| 13 | 5 - 20 | 4 | (Lange et al., 1998) | North Alaska |
| 39 | - | 1 | (Oechel and Collins, 1976) | North Alaska |
| 2 | - | 1 | (Uchida et al., 2006) | Svalbard |
| 7 | - | 1 | (Uchida et al., 2002) | Svalbard |
| 7 | 1 - 21 | 6 | (Koshurnikova, 2007) | West Siberia |
| 35 | 16 - 53 | 2 | (O'Connell et al., 2003) | BOREAS S |
| 50 | - | 1 | (O'Donnell et al., 2011) | Central Alaska |
| 78 | - | 1 | (Frolking et al., 1996) | Manitoba |
| 93 | - | 1 | (Benscoter and Vitt, 2007) | Alberta |
| 60 | - | 1 | (Skre and Oechel, 1979) | Central Alaska |
| 68 | - | 1 | (Hermle et al., 2010) | Quebec |
| 25 | - | 1 | (Trumbore and Harden, 1997) | BOREAS N |
| 27 (2 - 104) | | | | |

**Table 4.** Studies which measure surface coverage in the biomes "boreal forest" and "tundra". Median and range of all values listed in a study are shown, "#" stands for the number of values. The BOREAS N study area is located in Manitoba, Canada and BOREAS S is located in Saskatchewan, Canada (Gower et al., 1997). The last row shows median and range of the values in the first column.

| Median | Range | # | Study | Location |
|---|---|---|---|---|
| 0.8 | - | 1 | (Bisbee et al., 2001) | BOREAS S |
| 0.95 | - | 1 | (Swanson and Flanagan, 2001) | BOREAS S |
| 0.65 | 0.13 - 0.95 | 8 | (Bond-Lamberty and Gower, 2007) | BOREAS N |
| 0.93 | 0.77 - 0.99 | 13 | (Camill et al., 2001) | BOREAS N |
| 0.5 | - | 1 | (Street et al., 2012) | North Sweden |
| 0.6 | - | 1 | (Kolari et al., 2006) | South Finland |
| 0.5 | - | 1 | (Wahren et al., 2005) | Alaska |
| 0.57 | - | - | (Rapalee et al., 2001) | BOREAS region |
| 0.63 (0.5 - 0.95) | | | | |

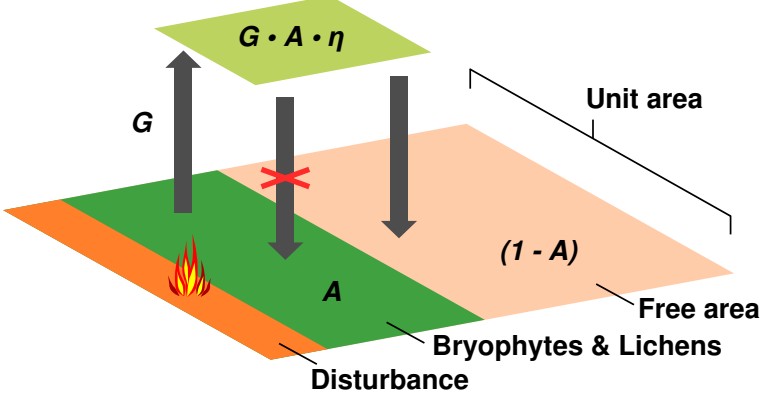

**Potential new bryophyte & lichen area**

**Figure 1.** Change in surface coverage $A$ of bryophytes and lichens. $A$ is reduced by disturbance and increases due to net growth $G$. New bryophyte and lichen cover per unit area is limited by the generating area $A$, the "expansion efficiency" $\eta$ and the available free area $(1 - A)$.

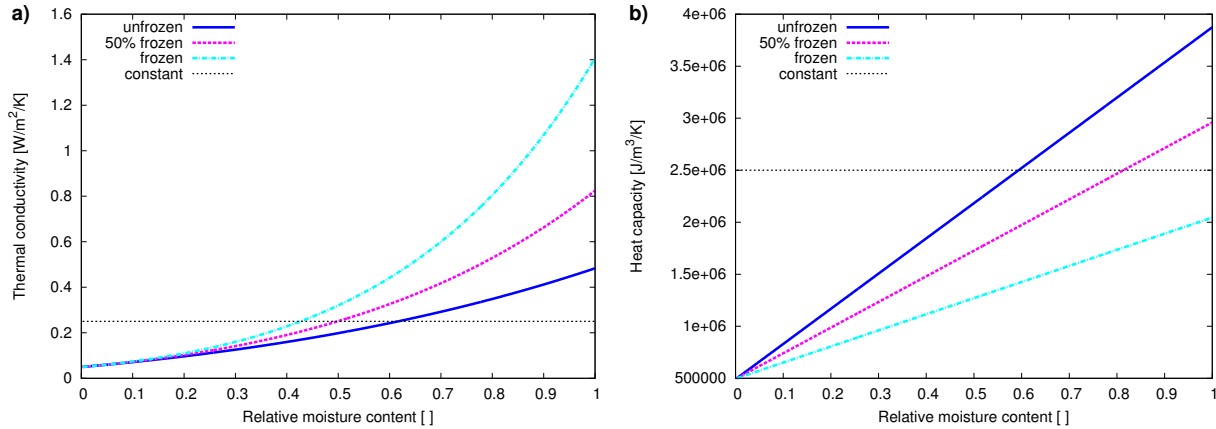

**Figure 2.** Dependence of a) thermal conductivity and b) heat capacity of the bryophyte and lichen cover on relative moisture content. The solid blue curves show the relation between thermal properties and moisture content for liquid water, the magenta dashed curves correspond to a mixture of 50 % liquid and 50 % frozen water and the light blue dashed-dotted curves correspond to frozen water. The black dotted lines show the thermal properties of the organic layer from the previous model version (Ekici et al., 2014) for comparison. They are constant since they do not depend on water content.

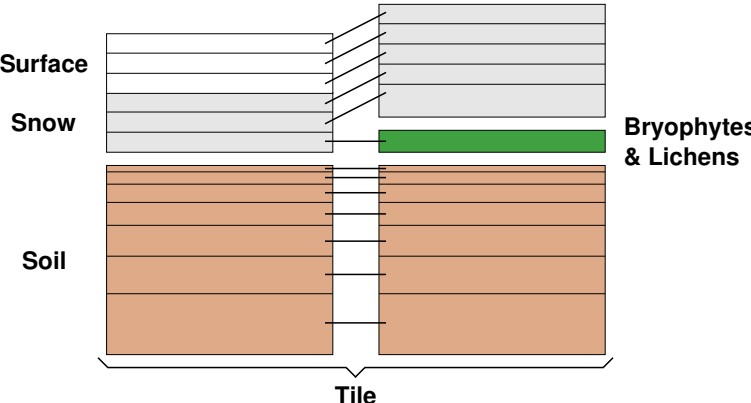

**Figure 3.** Averaging scheme for the vertical temperature profile of a tile: The left column corresponds to the part of the tile where no bryophyte and lichen layer is present, the right column corresponds to the part covered by bryophytes and lichens. The columns consist of 7 soil layers (brown), an optional bryophyte and lichen layer (green), up to 5 snow layers (grey) and several optional "empty" layers (white). The black horizontal bars connect layers which are averaged with regard to their temperature. The surface fractions of the two parts are used as weights for the averaging. The "empty" layers are assumed to have surface temperature.

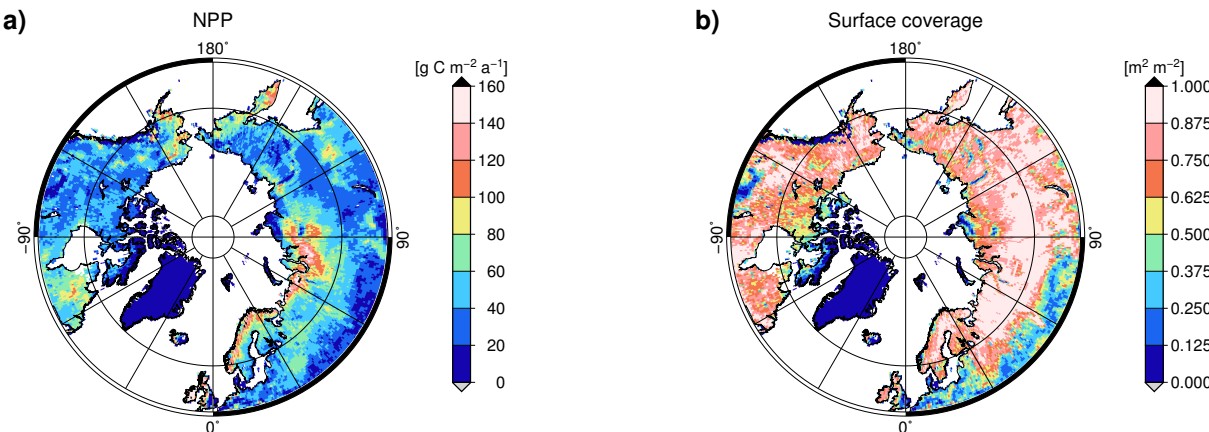

**Figure 4.** Large-scale patterns of a) NPP and b) surface coverage of bryophytes and lichens simulated in JSBACH. The values in the grid cells are calculated by averaging the last 15 years of a 1901 to 2010 "Dynamic" simulation. "g C" stands for "gram carbon".

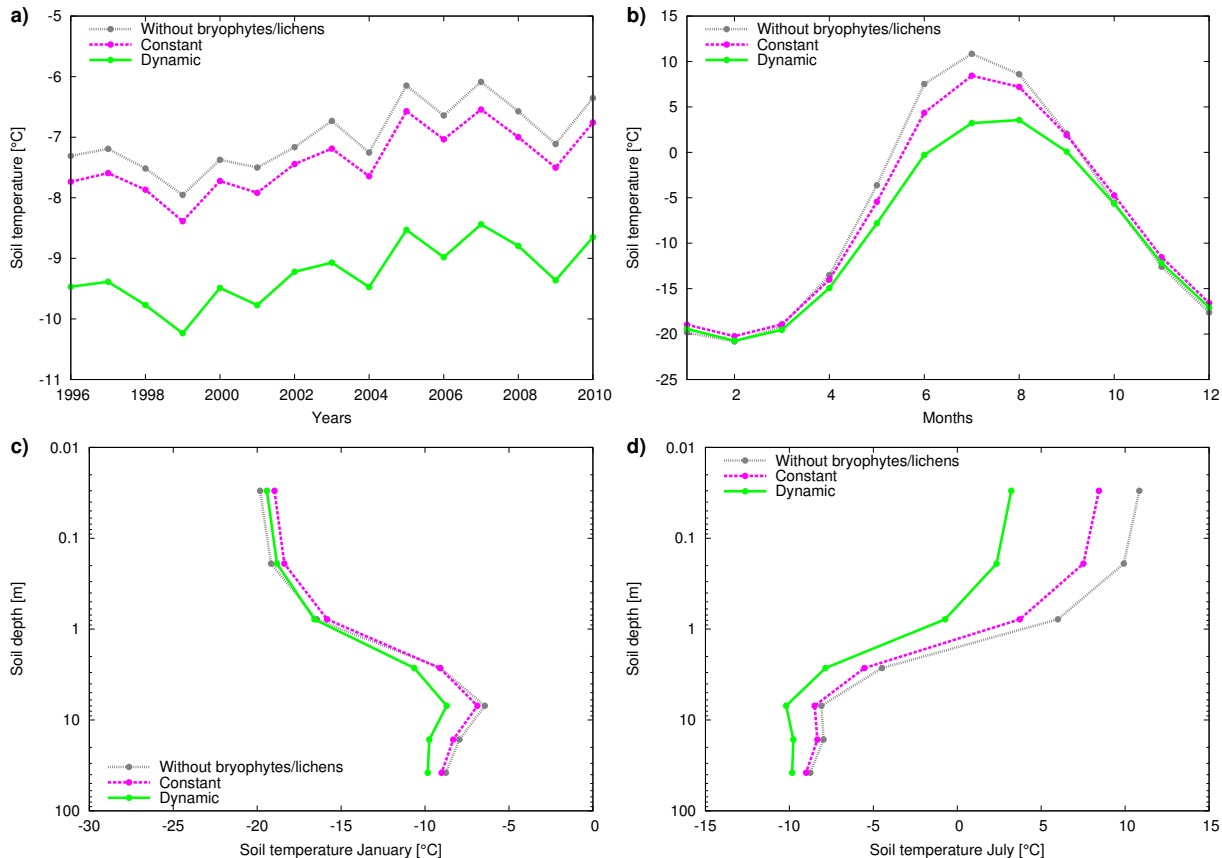

**Figure 5.** Temperature of the uppermost soil layer (a,b) and the whole simulated soil column (c,d), averaged over the study region for the biomes "boreal forest" and "tundra". The dots stand for a) annual average values, b) monthly average values, or c,d) values at the mid-depth of a soil layer in JSBACH. The mid-depths of the layers are at 0.03, 0.19, 0.78, 2.68, 6.98, 16.4 and 38.11 m depth. "Without" denotes a transient simulation from 1901 to 2010 without a bryophyte and lichen layer, "Constant" denotes an equivalent simulation, but including an organic layer with constant coverage and thermal properties and "Dynamic" stands for a bryophyte and lichen layer with dynamic coverage and thermal properties.

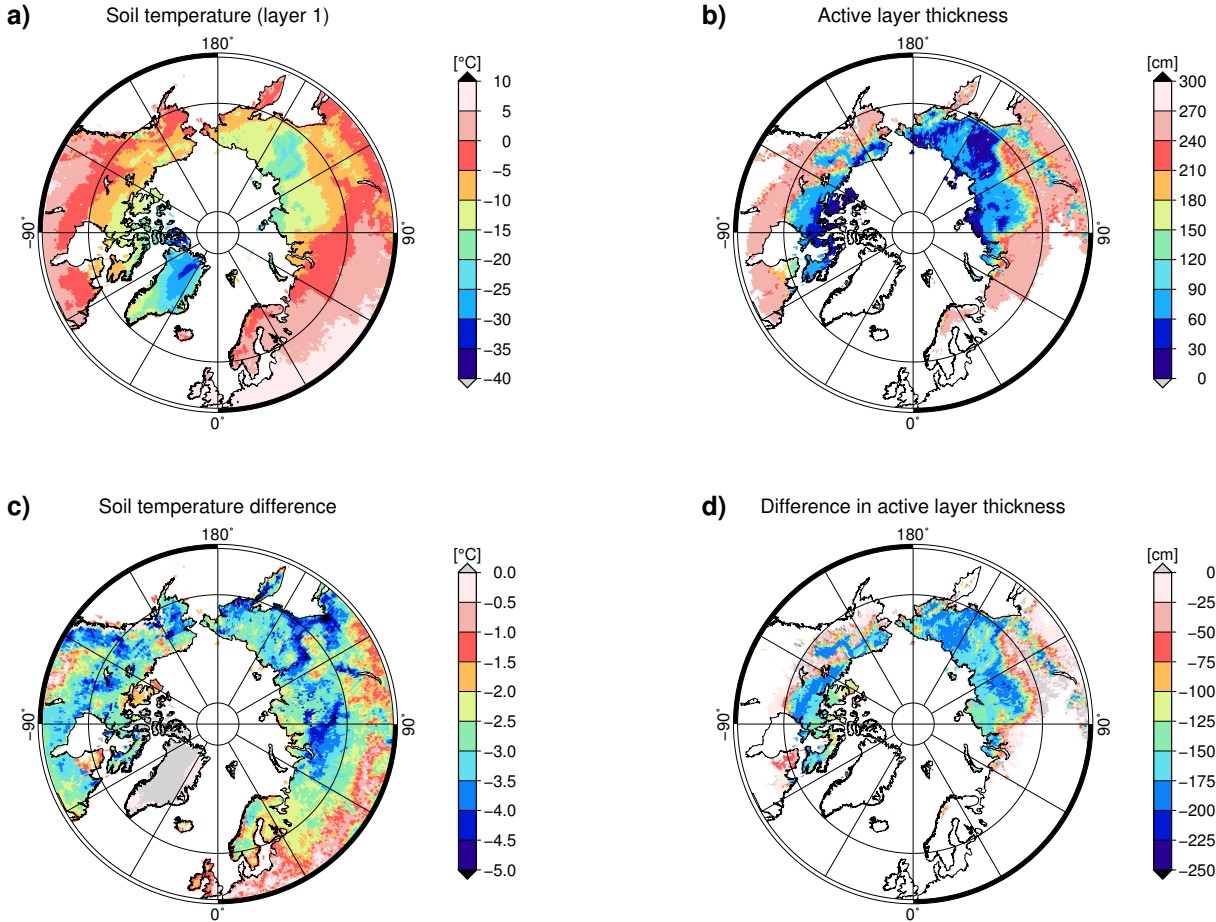

**Figure 6.** The maps display the spatial patterns of a) topsoil temperature, b) active layer thickness, c) the temperature difference in the uppermost soil layer and d) the difference in active layer thickness, based on average values over the last 15 years of transient simulations from 1901 to 2010. The differences are calculated by subtracting output of the "Without" simulation with no bryophyte and lichen layer from the "Dynamic" one.

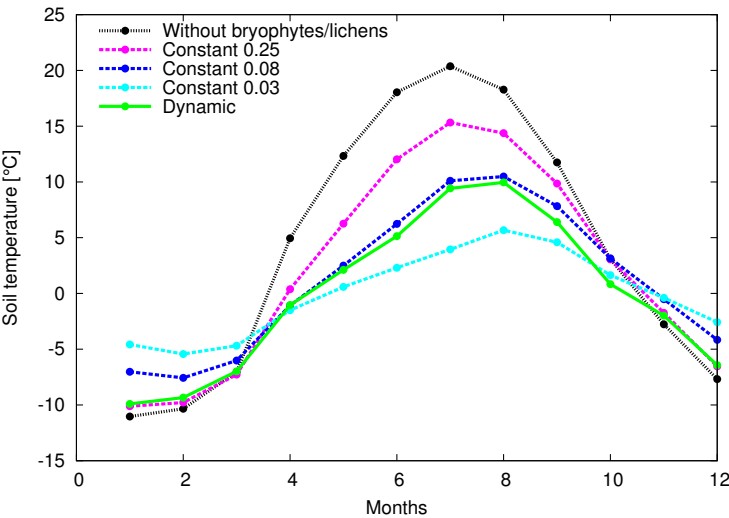

**Figure 7.** Temperature of the uppermost soil layer of a model grid cell at 55° 30' N, 98° 30' W, which is also used for the sensitivity analysis (Sect. A in the appendix). The dots stand for monthly average values for the period 1996 to 2010. "Without" denotes a transient simulation from 1901 to 2010 without a bryophyte and lichen layer. "Constant 0.25" denotes an equivalent simulation, but including an organic layer with constant coverage and thermal properties. The thermal conductivity of this layer is set to 0.25 $[\text{W K}^{-1}\,\text{m}^{-1}]$ as in Ekici et al. (2014). "Constant 0.08" corresponds to an organic layer calibrated to reproduce topsoil temperature under the bryophyte and lichen layer in summer. "Constant 0.03" corresponds to an organic layer calibrated to reproduce average annual topsoil temperature under the bryophyte and lichen layer. "Dynamic" stands for a bryophyte and lichen layer with dynamic coverage and thermal properties.

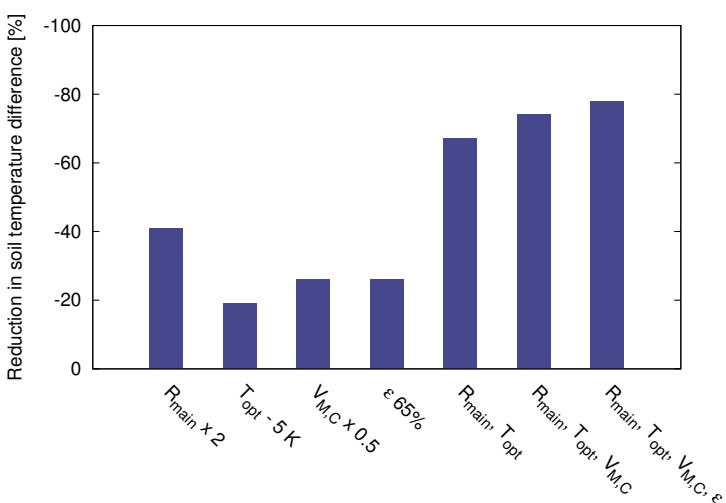

**Figure 8.** Effect of model parameters on simulated difference in topsoil temperature between the "Dynamic" and "Without" simulations. The individual effects of four varied parameters are shown, which significantly reduce the temperature difference. These are the specific maintenance respiration rate $R_{\mathrm{main}}$, the optimum temperature of photosynthesis, $T_{\mathrm{opt}}$, the molar carboxylation rate of Rubisco, $V_{\mathrm{M,C}}$ and the porosity $\epsilon$ of the bryophyte and lichen layer (see Tab. 2 for details). Also shown are the combined effects of $R_{\mathrm{main}}$ and $T_{\mathrm{opt}}$, $R_{\mathrm{main}}$, $T_{\mathrm{opt}}$ and $V_{\mathrm{M,C}}$, as well as $R_{\mathrm{main}}$, $T_{\mathrm{opt}}$, $V_{\mathrm{M,C}}$ and $\epsilon$. The effect on the temperature difference is shown in percentage of reduction.

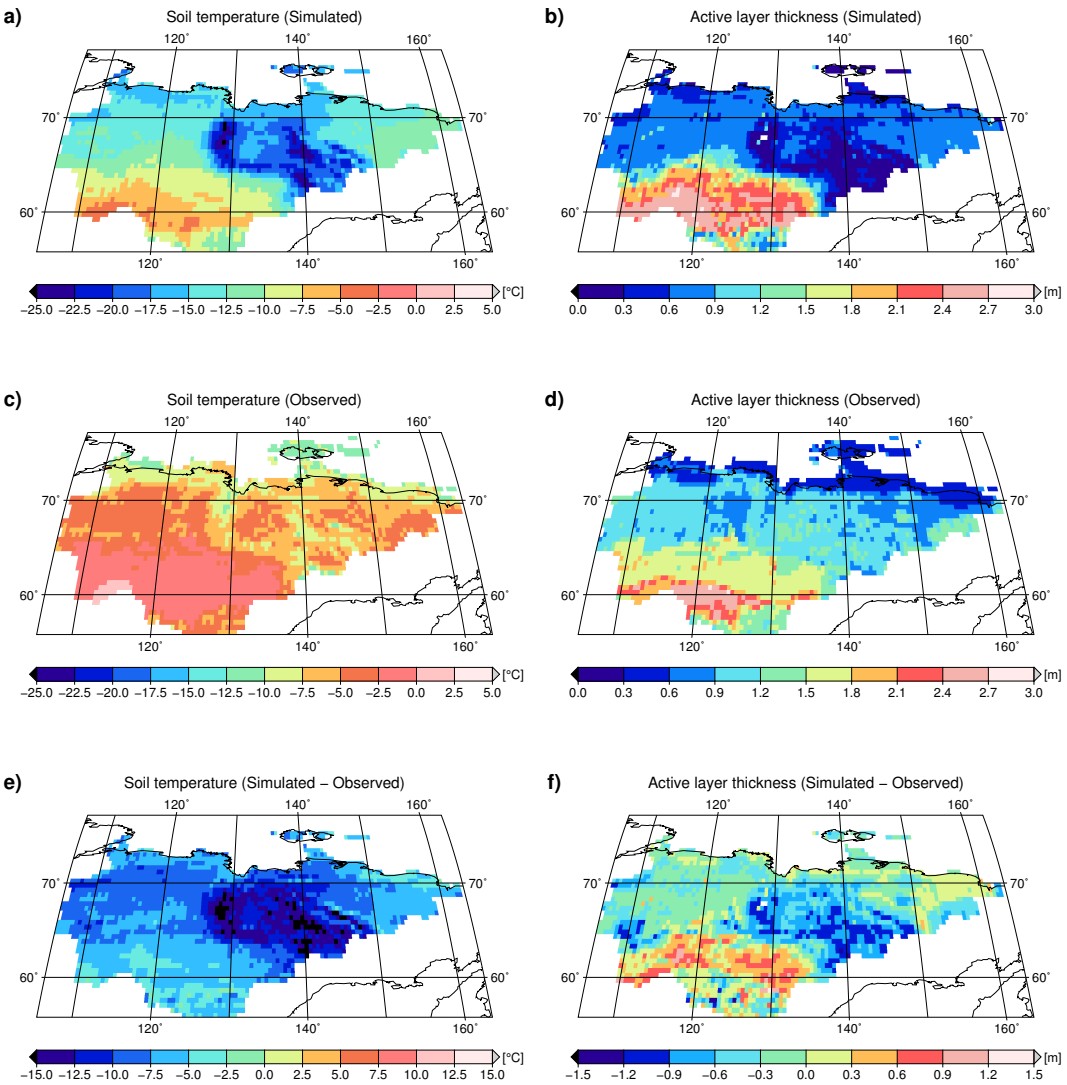

**Figure 9.** The maps show the spatial patterns of a) simulated subsoil temperature and b) active layer thickness, c) observed subsoil temperature and d) active layer thickness as well as the differences between simulation and observations regarding e) subsoil temperature and e) active layer thickness for the region of Yakutia. The maps a) and b) are based on average values over the last 15 years of a transient "Dynamic" simulation from 1901 to 2010. Subsoil temperature is calculated by averaging the temperatures of the third and fourth soil layer of JSBACH. The observation-based maps in c) and d) are taken from Beer et al. (2013).

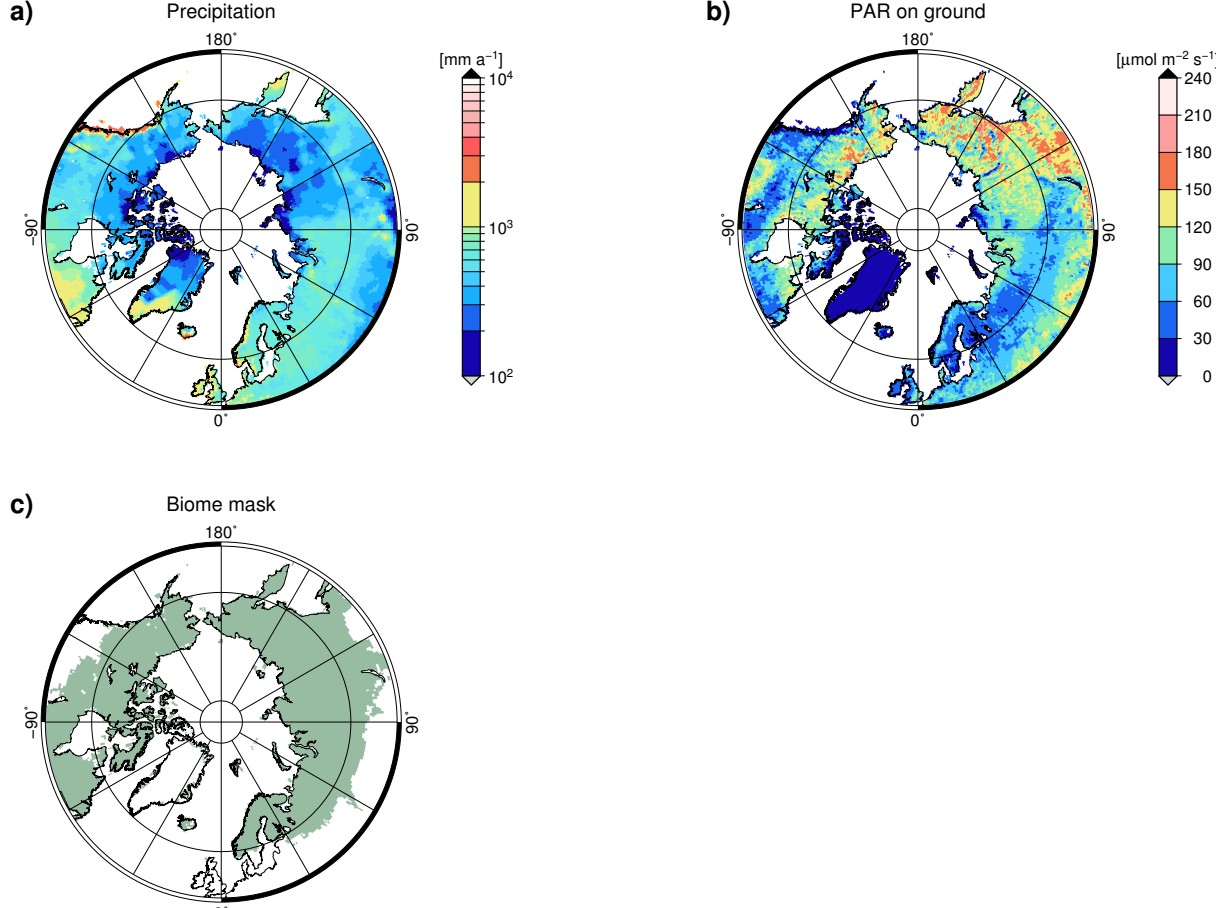

**Figure 10.** Additional model output, averaged over the last 15 years of a JSBACH transient simulation form 1901 to 2010. Spatial patterns of a) precipitation and b) photosynthetically active radiation (PAR) on the ground. Note that PAR on ground is a variable of LiBry in JSBACH, it is therefore only larger than zero where bryophytes and lichens exist in the model. c) shows the biome mask based on Olson et al. (2001) which is used to constrain our estimates to "boreal forest" and "tundra".