# Peer review of "Effects of bryophyte and lichen cover on permafrost soil temperature at large scale"

_The Cryosphere, 2015_

## Referee Comment (RC1) · Anonymous Referee #1 · 1 Mar 2016

General comments:

Bryophytes (mosses) and lichens are known to have an insulating function, thereby protecting the permafrost in the Arctic region, but this is not or poorly taken into account in earth system models. In this study a dynamic bryophyte and lichen layer has been added to the global land surface model JSBACH. Both the coverage and thermal properties of this new layer are dynamic as the water content is taken into account. As such this paper is a novel contribution within the scope of The Cryosphere. The authors have addressed the effect of the insulating bryophyte and lichen layer by comparing with model simulations without such a layer or a constant (non-dynamic) bryophyte and lichen layer. Outcome is a significant impact of the bryophyte and lichen layer on soil temperature and associated permafrost thawing depth in summer in tundra and boreal forest areas. In general, the study is well conducted and the manuscript

is well written, although some editing still needs to be done. Figures and tables are well designed. My main comment is that the manuscript is rather long for a modelling study with a limited scope. Perhaps move Model sensitivity and Model evaluation to Appendices (with summaries in the main text)?

Specific comments:

P.2, L.16-17: Therefore, ... please be more specific about what is expected for the insulating effect in the different seasons.

P.4, L.20: Suggestion: mortality instead of turnover.

P.5, L.4: Isn't G (net growth) per unit of time? What is the time step of the calculations?

P.5, L.7: The disturbance takes place once a year?

P.8, L.28: How large are the grid cells in the model?

P.10, L.3-6: It would be helpful to give these three configurations a name, e.g. Dynamic, Constant, and Without (bryophyte and lichen layer).

P.10, L.8: Suggestion: start a new section here, e.g. 2.3 Sensitivity analysis.

P.14, Fig. 5c: Suggestion: use a similar range of the x-axis scale, e.g. -30 to 0, for a proper comparison with panel d.

P.15, Fig. 6 legend: Is the difference Dynamic minus Without or Dynamic minus Constant bryophyte and lichen layer?

P.16, L.12-15: I do not understand how an increased NPP can lead to a reduced coverage and what porosity has to do with this. Equation 1 shows how coverage depends positively on net growth. And net growth is positively related to NPP.

P.16, L.16-20: Not necessary to discuss this, as effect is only small

P.17, L. 14-15: Is this because the chosen grid cell is a relatively warm grid cell?

[Figure]

P.18, Table 2 legend first line: Suggestion: temperature difference due to the addition of a dynamic bryophyte and lichen layer

Table 2: deltaT/deltaP is not explained in the legend and not used in the text

P.19 This section Model evaluation comes as a surprise: it was not announced in the Methods

Text comments:

Results are usually written in past tense. Please pay attention to this, particularly in the Discussion when you describe your study (which is done).

P.19, L.5: insert north in-between region and of

P.19, L.23: to a similar extent as

P.25, L.5: reducing effect of ... ground cover on ...

———————————————

---

## Referee Comment (RC2) · Anonymous Referee #2 · 8 Apr 2016

**General comments**

This manuscript builds on current understanding of the role of bryophytes and lichens in soil thermal regimes at northern high latitudes. Mosses and lichens often dominate the ground cover of northern ecosystems and play important roles in a variety of ecosystem processes including nutrient and precipitation interception, nutrient sequestration away from vascular plants, and soil cooling. The goal of this study is to integrate a process-based vegetation model that includes nonvascular plant growth into the land surface model JSBACH. The authors conduct a model experiment comparing model output between treatments with and without the addition of this bryophyte/lichen cover. While the results all support general expectations based on our current knowledge, I think this type of model coupling is an important step forward in more accurate representation of permafrost thermal regimes. My only major comment on this study is that I would recommend that the authors be more clear in the abstract and introduction about what ecosystems are being represented. Clearly, moss impacts on soil nutrient cycling and thermal regimes have been most well studied in boreal and subarctic peatlands. It is not until the very end of this paper that the reader learns that peatlands are not included in this model framework. This needs to be articulated much earlier in the paper, and defended given that many of the studies used for estimating parameters (Tables 3 and 4) include peatlands.

**Specific comments**

1. Page 1, line 23: But mosses are being outcompeted by shrubs in many northern systems, which can often involve an increase in productivity.

2. Page 2, lines 13-17: I certainly agree. This in part is why Bona et al. (2013) recommend that carbon cycling models parameterize feather mosses separately from Sphagnum mosses. [Bona et al. 2013. Ecosystems16: 1071]

3. Page 3, lines 7-9. The authors describe three model scenarios here: dynamic nonvascular layer, static nonvascular layer, no nonvascular layer. Later in the manuscript, particularly in figure and table legends, it would be helpful to make sure that it is clear which scenarios are being depicted. I found it difficult to interpret in some figure legends what scenarios were being portrayed. If the authors are representing differences between two scenarios, is it always between the dynamic versus no nonvascular scenarios?

4. Page 4, line 10. While nonvascular plants lack a vascular system (though some species have poorly developed vascular tissue), they absolutely do exert controls on evaporation. This is achieved not only through individual level traits, but also community level traits. Thus, it does not seem reasonable to me that the moss evaporation rate would be similar to an interception reservoir.

5. Page 10, line 17. To be consistent with ecological studies, it would be helpful to refer to the depth of the living moss layer or the depth of the living + dead moss mat. It is notoriously difficult to separate live moss tissue from older dead

organic material in many boreal and arctic ecosystems.  While this would have little implications for C uptake rates, it would influence thermal properties.

6.  All empirical studies find that the thickness of the moss mat is one of the most important determinants of ground thermal regimes.  Yet this study includes a static depth of 4.5 cm.  I would like to see the sensitivity analyses include a greater range of variation in moss thickness.  Different moss functional groups vary immensely in interactions between thickness and porosity, and this is going to govern influence thermal properties.

7.  Figure 4.  Which of the three model scenarios is being portrayed here?  I assume this is the dynamic nonvascular scenario.

8.  Page 13, line 12:  There is no boreal forest biome.  There are boreal forests within the boreal biome.  The boreal biome includes boreal peatlands, boreal forests, etc.

9.  Table 1.  This table confuses me.  Aren't all the rows representing differences between scenarios?  Why is "difference" only listed in the last row?

10. Table 2.  I think visualizing some of these results graphically might be more effective.  For the parameters found to be most important (sensitive) through this approach, I think it would be particularly interesting to run additional analyses to explore interactions between parameters.

11. Table 3.  This does not appear to be a comprehensive list of studies that have measured moss NPP in boreal or arctic regions.  There are many studies in the literature not included here.  It would be helpful to articulate why studies were or were not selected for inclusion in this table.  Also, some of the studies listed in this table measured moss NPP in peatlands.  I do not think these should be included, and it should be articulated why from the start of this paper.

12. Table 4.  Data from New Hampshire should not be included here.

13. Figure 7.  More information needs to be included in figure legends to make these "stand-alone".  What are the observed data?  This information is in the main text, but needs to be briefly described in the legend as well.

---

## Author Comment (AC1) · 6 May 2016

**Effects of bryophyte and lichen cover on permafrost soil temperature at large scale**

P. Porada [*1,2], A. Ekici[3], and C. Beer[1,2]

[1]Department of Environmental Science and Analytical Chemistry, Stockholm University,
Svante Arrhenius väg 8, SE-11418 Stockholm
[2]Bolin Centre for Climate Research, Stockholm University, 10691 Stockholm
[3]Earth System Sciences, Laver Building, University of Exeter, Exeter, UK

May 6, 2016

**Summary**

We thank the reviewers for their comments which helped to improve our manuscript. In the revised manuscript, we account for all points raised by the reviewers. In particular, we shifted the model sensitivity analysis and evaluation to appendices, following reviewer #1 and we removed studies from peatlands from our database used to evaluate the model, as requested by reviewer #2. Moreover, we point out that the JSBACH model is not designed to represent peatlands and that our study focuses on upland soils.

**Response to the comments of reviewer #1**

We show the reviewer's comments in italic text, while our responses are formatted as standard text.

*General comments: Bryophytes (mosses) and lichens are known to have an insulating function, thereby protecting the permafrost in the Arctic region, but this is not or poorly taken into account in earth system models. In this study a dynamic bryophyte and lichen layer has been added to the global land surface model JSBACH. Both the coverage and thermal properties of this new layer are dynamic as the water content is taken into account. As such this paper is a novel contribution within the scope of The Cryosphere. The authors have addressed the effect of the insulating bryophyte and lichen layer by comparing with model simulations without such a layer or a constant (non-dynamic) bryophyte and lichen layer. Outcome is a significant impact of the bryophyte and lichen layer on soil temperature and associated permafrost thawing depth in summer in tundra and boreal forest areas. In general, the study is well conducted and the manuscript is*
* * *
[*]Philipp.Porada@aces.su.se; Corresponding author

*well written, although some editing still needs to be done. Figures and tables are well designed. My main comment is that the manuscript is rather long for a modelling study with a limited scope. Perhaps move Model sensitivity and Model evaluation to Appendices (with summaries in the main text)?*

We shifted sensitivity analysis and evaluation to appendices and provide short summaries in the methods and results section.

*Specific comments:*

*P.2, L.16-17: Therefore, ... please be more specific about what is expected for the insulating effect in the different seasons.*

We extended the text by the following: "Therefore, a large and important seasonality of the insulating effect is expected: In winter and, to a lesser extent, in autumn and spring, the insulating effect of bryophytes and lichens will be small due to the high thermal conductivity of ice and water. In summer, however, a dry bryophyte and lichen cover may strongly insulate the ground, leading to a net cooling effect for the whole year."

*P.4, L.20: Suggestion: mortality instead of turnover.*

We replaced this by "Turnover of biomass associated with mortality" (P.4, L.33)

*P.5, L.4: Isnt G (net growth) per unit of time? What is the time step of the calculations?*

Yes, G is per unit of time. However, in the model a change in surface cover takes place only once a month. Therefore, G is accumulated over a month. We clarified this in the text (P.5, L.7). The time step of the simulation is 30 minutes. This is described in the section "Model setup" (P.10, L.1).

*P.5, L.7: The disturbance takes place once a year?*

No, the disturbance takes place once a month, the disturbance interval is converted accordingly. We clarified this in the text (P.5, L.7).

*P.8, L.28: How large are the grid cells in the model?*

The size of the grid cells depends on the latitude considered. We added an example for the size of the grid cells to the text (approximately 30 x 30 km at 60° N) (P.9, L.8).

*P.10, L.3-6: It would be helpful to give these three configurations a name, e.g. Dynamic, Constant, and Without (bryophyte and lichen layer).*

We added the following sentence to the text (P.10, L.23):"These 3 simulations will be referred to as "Dynamic", "Constant" and "Without" througout the text" and we changed the manuscript accordingly.

*P.10, L.8: Suggestion: start a new section here, e.g. 2.3 Sensitivity analysis.*

We moved instead the sections sensitivity analysis and model evaluation to appendices.

*P.14, Fig. 5c: Suggestion: use a similar range of the x-axis scale, e.g. -30 to 0, for a proper comparison with panel d.*

We adjusted the x-axis range accordingly.

*P.15, Fig. 6 legend: Is the difference Dynamic minus Without or Dynamic minus Constant bryophyte and lichen layer?*

The difference is Dynamic minus Without, we made this more clear in the legend.

*P.16, L.12-15: I do not understand how an increased NPP can lead to a reduced coverage and what porosity has to do with this. Equation 1 shows how coverage depends positively on net growth. And net growth is positively related to NPP.*

Net growth is indeed positively related to NPP, but NPP in [g m$^{-2}$ of cover] per month is multiplied by specific area [m$^2$ of cover g$^{-1}$] to obtain the net growth of new cover per cover per month. To clarify this, we changed the text to (P.20, L.31): "Interestingly, a reduction in porosity leads to a reduced coverage in spite of slightly increased NPP. The reason for this is that lower porosity results in a smaller specific area of the bryophyte and lichen PFT. Since net growth of cover (Eq. 1) is calculated by multiplying NPP with specific area, the effect of reduced specific area may overrule an increase in NPP. This leads to less cover expansion.".

*P.16, L.16-20: Not necessary to discuss this, as effect is only small*

We did not remove this part since reviewer #2 was interested in further analysis of the variation in layer thickness.

*P.17, L. 14-15: Is this because the chosen grid cell is a relatively warm grid cell?*

To clarify, we extended the text by the following (P.21, L.28): "The reason for the asymmetric response of NPP to $T_{opt}$ is the exponential dependence of respiration on the difference between surface temperature and $T_{opt}$. Since $T_{opt}$ is already relatively high, a further increase leads only to a small reduction in respiration. Lowering $T_{opt}$, however, results in a strong increase in respiration and, consequently, reduced NPP."

*P.18, Table 2 legend first line: Suggestion: temperature difference due to the addition of a dynamic bryophyte and lichen layer*

We added this to the text.

*Table 2: deltaT/deltaP is not explained in the legend and not used in the text*

We added to the legend: "$\Delta T/\Delta P$ is the percentual change in temperature difference ($\Delta T$) due to a change in a model parameter ($\Delta P$), compared to the control run."

*P.19 This section Model evaluation comes as a surprise: it was not announced in the Methods*

We added a short summary of our method of evaluation to the section Model setup (P. 10, L.30).

*Text comments:*

*Results are usually written in past tense. Please pay attention to this, particularly in the Discussion when you describe your study (which is done).*

We changed several sentences in the Discussion to past tense. However, we left general statements in present tense, such as "The spatial pattern ... can be structured ..." or "JSBACH is designed to ..." or "This is a substantial effect ...". We do not want to give the impression that the respective statement was only valid in the past. We left the Results section in present tense to avoid frequent jumping between present and past tense, which would make it harder to read. However, we can change this on request of the editor.

*P.19, L.5: insert north in-between region and of*
*P.19, L.23: to a similar extent as*
*P.25, L.5: reducing effect of ... ground cover on ...*

We changed the manuscript text accordingly.

**Response to the comments of reviewer #2**

*General comments: This manuscript builds on current understanding of the role of bryophytes and lichens in soil thermal regimes at northern high latitudes. Mosses and lichens often dominate the ground cover of northern ecosystems and play important roles in a variety of ecosystem processes including nutrient and precipitation interception, nutrient sequestration away from vascular plants, and soil cooling. The goal of this study is*

*to integrate a process-based vegetation model that includes nonvascular plant growth into the land surface model JSBACH. The authors conduct a model experiment comparing model output between treatments with and without the addition of this bryophyte/lichen cover. While the results all support general expectations based on our current knowledge, I think this type of model coupling is an important step forward in more accurate representation of permafrost thermal regimes. My only major comment on this study is that I would recommend that the authors be more clear in the abstract and introduction about what ecosystems are being represented. Clearly, moss impacts on soil nutrient cycling and thermal regimes have been most well studied in boreal and subarctic peatlands. It is not until the very end of this paper that the reader learns that peatlands are not included in this model framework. This needs to be articulated much earlier in the paper, and defended given that many of the studies used for estimating parameters (Tables 3 and 4) include peatlands.*

We point out in the revised abstract that JSBACH does not include bryophyte and lichen growth in peatlands. Furthermore, we added the following text to the introduction (P.3, L.13)): "Since JSBACH does not include a scheme for wetland hydrology, the formation of peatlands cannot be simulated. Hence, LiBry in JSBACH mainly represents bryophyte and lichen growth on upland forest floor sites."

We removed studies which measure surface coverage and NPP in peatlands from our Tables 3 and 4. We also excluded data from peatlands from studies including sites with different hydrological conditions.

*Specific comments*

*1. Page 1, line 23: But mosses are being outcompeted by shrubs in many northern systems, which can often involve an increase in productivity.*

We extended the sentence to (P.1, L.23): "This might lead to an increased release of carbon from the soil to the atmosphere which would accelerate global warming [Schuur et al., 2015], although this may be partially compensated by increased productivity of e.g. shrubs in northern ecosystems."

*2. Page 2, lines 13-17: I certainly agree. This in part is why Bona et al. (2013) recommend that carbon cycling models parameterize feather mosses separately from Sphagnum mosses. [Bona et al. 2013. Ecosystems16: 1071]*

We added the following to the text (P.2, L.21): "The relation between thermal properties of bryophytes and lichens and their water saturation is very similar for a large range of different species [Soudzilovskaia et al., 2013]. However, the species show considerable differences in water holding capacity and other properties which affect their water status. Therefore, different species may exhibit differing water saturation under similar climatic conditions."

We also added a new paragraph to the discussion where we point out potential effects

of using only one PFT to represent bryophytes and lichens in JSBACH (P.17, L.23): "The bryophyte and lichen layer is represented by one single PFT in JSBACH, for reasons of consistency with vascular vegetation in JSBACH and also for computational efficiency. However, this lack of diversity may have consequences for the estimated effect of the bryophyte and lichen layer on soil temperature. Given similar climatic conditions, bryophyte and lichen species may differ in their degree of water saturation and, consequently, in their thermal properties. If large regions differ in their dominant bryophyte or lichen species, this may affect our estimated patterns of soil temperature and active layer thickness. Additionally, bryophyte and lichen species differ in their thickness, while thickness within a species is relatively constant [Soudzilovskaia et al., 2013]. Since the bryophyte and lichen PFT in JSBACH has a constant thickness, we may underestimate further spatial effects on the soil thermal regime."

*3. Page 3, lines 7-9. The authors describe three model scenarios here: dynamic nonvascular layer, static nonvascular layer, no nonvascular layer. Later in the manuscript, particularly in figure and table legends, it would be helpful to make sure that it is clear which scenarios are being depicted. I found it difficult to interpret in some figure legends what scenarios were being portrayed. If the authors are representing differences between two scenarios, is it always between the dynamic versus no nonvascular scenarios?*

In the revised manuscript, we distinguish the three scenarios by the names "Dynamic", "Constant" and "Without", which we use througout the text and the table and figure legends. This should clarify which scenarios are shown in the respective figure. The differences are always between the "Dynamic" and the "Without" simulations, except for Fig. 5, where all three scenarios are shown.

*4. Page 4, line 10. While nonvascular plants lack a vascular system (though some species have poorly developed vascular tissue), they absolutely do exert controls on evaporation. This is achieved not only through individual level traits, but also community level traits. Thus, it does not seem reasonable to me that the moss evaporation rate would be similar to an interception reservoir.*

It is true that bryophytes and lichens are able to exert a certain control on evaporation, however they are much more limited in this respect than vascular plants [Nash III, 1996, Proctor, 2000]. The original version of LiBry considers reduced evaporation due to morphological traits. JSBACH, however, uses a relatively inflexible scheme for evapotranspiration, which makes it impractical to add fluxes of water from the surface to the atmosphere to the already implemented ones. These are evaporation from bare soil and the interception reservoir and transpiration. The interception reservoir includes vegetation surfaces on the ground as well as in the canopy. To clarify this, we extended the manuscript by the following text (P.4, L.20): "...Since it is impractical to modify this scheme, we approximate evaporation from bryophytes and lichens by evaporation from the interception reservoir. This potentially neglects the morphological control of the organisms on evaporation. However, free evaporation is more suitable than transpiration

by vascular plants to describe water loss of bryophytes and lichens . . . ".

*5. Page 10, line 17. To be consistent with ecological studies, it would be helpful to refer to the depth of the living moss layer or the depth of the living + dead moss mat. It is notoriously difficult to separate live moss tissue from older dead organic material in many boreal and arctic ecosystems. While this would have little implications for C uptake rates, it would influence thermal properties.*

To be more clear in this respect, we added the following to the text (P.6, L.14): " . . . we set thickness to 4.5 cm, which corresponds to the median of the measured values. The measurements are based on green and undecomposed brown tissue of bryophyte mats". Moreover, we added (P.19, L.11): "The thickness $z$ of the simulated bryophyte and lichen layer is set to 4.5 cm in the model. It is defined as the undecomposed living and dead parts of a bryophyte or lichen mat".

*6. All empirical studies find that the thickness of the moss mat is one of the most important determinants of ground thermal regimes. Yet this study includes a static depth of 4.5 cm. I would like to see the sensitivity analyses include a greater range of variation in moss thickness. Different moss functional groups vary immensely in interactions between thickness and porosity, and this is going to govern influence thermal properties.*

We extended the range of variation in thickness of the bryophyte and lichen layer and changed the text in the section Sensitivity analysis accordingly (P.19, L.14): "Here, we vary $z$ from 1 to 10 cm, which is a slightly larger range than determined by Soudzilovskaia et al. [2013] for 18 bryophyte species. We do not test extreme values of $z$, since we are interested in bryophyte and lichen species that are actually able to form a macroscopic ground cover".

    With these new values for $z$, we obtain a larger change in soil temperature difference: While the control run shows a difference of -2.7 K between "Dynamic" and "Without", the lower bound for $z$, 10 mm, shows -3.0 K and the upper bound, 100 mm, shows -2.4 K. Thus the percentual change in soil temperature difference now amounts to +11 % and -11 %, respectively. Before, it was $\pm\,4\,\%$ We extended the text for clarification (P.21, L.8): " . . . This means that the reducing effect of low moisture content on thermal conductivity of the bryophyte and lichen layer overrules the increasing effect of lower thickness in the model."

    We added a new paragraph to the discussion where we discuss the potential consequences of representing bryophytes and lichens in JSBACH by one PFT with a constant thickness and constant hydrological properties (P.17, L.23, see above).

*7. Figure 4. Which of the three model scenarios is being portrayed here? I assume this is the dynamic nonvascular scenario.*

Yes, Fig. 4 shows the dynamic scenario. We added this to the legend.

*8. Page 13, line 12: There is no boreal forest biome. There are boreal forests within the boreal biome. The boreal biome includes boreal peatlands, boreal forests, etc.*

We use the definition of Olson et al. [2001], who distinguish 14 biomes, and who use the names "boreal forest" and "tundra". The advantage of this classification is that it comes with a digital map which we can use to mask our study region. If we used the Walter classification, which contains a "boreal" biome, we would also have to change the biome "tundra" into "polar" for consistency. To make this more clear, we added a second reference for our biome classification to the text (P.23, L.10).

*9. Table 1. This table confuses me. Aren't all the rows representing differences between scenarios? Why is "difference" only listed in the last row?*

The first two rows of Tab. 1 do not show differences, but estimates based on the "Dynamic" simulation, since there is no NPP and surface coverage of lichens and bryophytes in the "Without" simulation. We clarify this by changing the legend to: "Net primary productivity (NPP) and surface coverage of the simulated dynamic bryophyte and lichen layer, as well as differences between the "Dynamic" and "Without" simulations with regard to topsoil temperature and active layer thickness...".

*10. Table 2. I think visualizing some of these results graphically might be more effective. For the parameters found to be most important (sensitive) through this approach, I think it would be particularly interesting to run additional analyses to explore interactions between parameters.*

We added a new figure to the revised manuscript (Fig. 7, the Yakutia plot is now Fig. 8). The figure shows the percentual reducing effect of four varied parameters on the difference in soil temperature between the "Dynamic" and "Without" scenario. It furthermore shows how combinations of these four parameters lead to increased reduction in temperature difference. We added the corresponding information to the sections Results and Sensitivity analysis (P.15, L.11; P.20, L.27; P.22, L.2).

*11. Table 3. This does not appear to be a comprehensive list of studies that have measured moss NPP in boreal or arctic regions. There are many studies in the literature not included here. It would be helpful to articulate why studies were or were not selected for inclusion in this table. Also, some of the studies listed in this table measured moss NPP in peatlands. I do not think these should be included, and it should be articulated why from the start of this paper.*

We removed NPP values measured in peatlands from Tab. 3. We furthermore added several studies which provide measurements of NPP to Tab. 3. In the appendix, we explain in more detail how we created the list of NPP measurements (P.23, L.6): "...Therefore, we compile a list of available small-scale observations from various studies, which are largely taken from overviews in Turetsky et al. [2010] and Bona et al. [2013]. We do not

attempt to create a comprehensive review of measurements of surface coverage and NPP in high-latitude regions. However, our list is sufficient to establish "characteristic" values of surface coverage and NPP for the region north of 50° N. Thereby, we constrain our analysis to the biomes "boreal forest" and "tundra" (after Olson et al. [2001], see also Fig. 8 c)). Since JSBACH is not primariy designed to simulate peatlands, we exclude studies which measure bryophyte NPP in peatlands. Several studies measure NPP both on peatland and upland sites. In this case, we do not consider values from peatlands. In many cases, the original studies provide NPP measurements in grams of biomass. We convert these values to grams of carbon using a factor of 0.45 [Bauer et al., 2009]."

*12. Table 4. Data from New Hampshire should not be included here.*

The data from New Hampshire are from an alpine tundra, which is why we thought they are suitable for comparison. To avoid misunderstandings, we excluded the data in the revised manuscript.

*13. Figure 7. More information needs to be included in figure legends to make these "stand-alone". What are the observed data? This information is in the main text, but needs to be briefly described in the legend as well.*

We added a reference for the observation-based maps to the figure legend.

[revised manuscript text omitted]

---

## Author Response (AR2)

**Effects of bryophyte and lichen cover on permafrost soil temperature at large scale**

P. Porada [*,1,2], A. Ekici[3,4], and C. Beer[1,2]

[1]Department of Environmental Science and Analytical Chemistry, Stockholm University,
Svante Arrhenius väg 8, SE-11418 Stockholm
[2]Bolin Centre for Climate Research, Stockholm University, 10691 Stockholm
[3]Earth System Sciences, Laver Building, University of Exeter, Exeter, UK
[4]Uni Research Climate, Bjerknes Centre for Climate Research, Bergen, Norway

September 15, 2016

**Response to the comments of reviewer #3**

We thank the reviewer for thoughtful comments which helped to improve our manuscript. In the revised manuscript, we account for all points raised by the reviewer. We discuss the differences between constant and dynamic parameterizations of an organic surface layer in more detail and we clarify better how changes in surface coverage are calculated in the model.

We show the reviewer's comments in italic text, while our responses are formatted as standard text.

*This is a very well written, clear and concise paper. The development work it reports on is well thought through. Choices made during the development stage are well justified. I only have some minor comments and suggestions, given below.*

*General comments*

*It is unclear to me which are the parameter values chosen in the "constant" simulation. Maybe they were given somewhere, but it did not find this even after looking twice. These values should be stated more clearly, as it would help interpreting the results.*

We added the following to our manuscript (P. 7, L. 29): "The thermal conductivity of this layer is set to 0.25 [$\mathrm{W\,K^{-1}\,m^{-1}}$] and its heat capacity is set to 2.5E6 [$\mathrm{J\,m^{-3}\,K^{-1}}$]. These values are shown in Fig. 2 for comparison."
* * *
[*]Philipp.Porada@aces.su.se; Corresponding author

*I would have liked to see a more detailed analysis of seasonal effects, in particular the difference between summer and the shoulder seasons (which might be a bit more humid?). As the paper is now, it is not shown convincingly that the dynamic hydric state of bryophytes (and its effect on soil temperature) is really required, i.e. that this effect cannot be taken into account by simply tweaking the average ("constant") heat conductivity and heat capacity.*

We extended our revised manuscript accordingly and discuss the differences between a constant and a dynamic parameterization in more detail (P. 13, L. 17): "Alternatively to introducing a bryophyte and lichen layer with dynamic thermal properties, one might ask if calibrating a constant organic layer would also be sufficient to simulate the insulating effect. In Fig. 7 we show that there are qualitative differences between varying thermal conductivity of a constant organic layer and the dynamic bryophyte and lichen layer: If constant thermal conductivity is calibrated to reproduce soil temperature under a bryophyte and lichen layer in summer, the temperature in winter and, consequently, annual average temperature is too warm. To simulate an average annual soil temperature similar to that under the bryophyte and lichen layer, the constant thermal conductivity has to be strongly reduced, since the warming effect in winter has to be compensated. This leads to an unrealistically weak seasonal amplitude of soil temperature.

We want to point out that the bryophyte and lichen layer should not be confused with an organic soil layer from a modelling perspective. So far, JSBACH does not include an organic soil layer. Differences between these two types of layers are the relatively low thickness of the bryophyte and lichen layer compared to the organic layer, which can be several tens of centimeters thick due to the accumulation of peat. This means that the organic layer may have a significantly higher heat capacity when saturated with water or ice. Therefore, the reducing effect of high thermal conductivity on insulation can be counteracted by the large amount of energy required for a temperature change of the layer, which increases insulation. Furthermore, the higher residual water content of the peat layer compared to the bryophyte and lichen layer may result in a lower variability of thermal conductivity. These two differences may explain why e.g. Atchley et al. [2016] find a relatively small influence of water content on the insulating effect at the surface, since their model approach includes an organic layer."

[Figure]

**Figure 7:** Temperature of the uppermost soil layer of a model grid cell at 55° 30' N, 98° 30' W, which is also used for the sensitivity analysis (Sect. A in the appendix). The dots stand for monthly average values for the period 1996 to 2010. "Without" denotes a transient simulation from 1901 to 2010 without a bryophyte and lichen layer. "Constant 0.25" denotes an equivalent simulation, but including an organic layer with constant coverage and thermal properties. The thermal conductivity of this layer is set to 0.25 $[\mathrm{W\,K^{-1}\,m^{-1}}]$ as in Ekici et al. [2014]. "Constant 0.08" corresponds to an organic layer calibrated to reproduce topsoil temperature under the bryophyte and lichen layer in summer. "Constant 0.03" corresponds to an organic layer calibrated to reproduce average annual topsoil temperature under the bryophyte and lichen layer. "Dynamic" stands for a bryophyte and lichen layer with dynamic coverage and thermal properties.

*The bryophytes seem to have little effect in winter, which makes sense. Is bryophyte compaction by snow taken into account? (Probably not, and probably it wouldn't make any additional difference because snow insulation is strong, but it would be good to know).*

We added the following to our manuscript (P. 8, L. 31): "Note that compaction of the bryophyte and lichen layer by snow is not considered here, since the effect of this process on overall thermal conductivity are probably small."

*The discussion about the differences between mosses and lichen could be a little bit more detailed, in terms of how their properties change with general humidity etc.*

We extended the discussion by the following (P. 15, L. 14): "Lichens, for instance, have in many cases a lower porosity and water storage capacity than bryophytes, particularly mosses [Schulze and Caldwell, 2012]. This means that lichens may be less sensitive to water content than bryophytes regarding their thermal properties. However, both groups include species which do not match this pattern."

*Specific comments*

*Quite often two sentences are contracted to a single one, only a comma separating the two parts. Examples: P1, L7: "The model simulates .. on upland sites, wetlands are not included". Or:P5, L19: "D is the fraction of cover lost due to disturbance, it is set to 0.00083 per month.". It would be better to have two separate sentences in these cases.*

We split the respective sentences into two separate ones.

*P1, L10: What is the "study region" mentioned here?*

We changed "study region" to "region north of 50° N"

*P2, L26: Indeed in ORCHIDEE there is no bryophyte PFT but Koven et al. (GRL, 2009) do take into account soil thermal insulation but an organic layer.*

We included this in our text (P. 2, L. 27): " Alternatively, the insulating effect is attributed solely to the carbon-rich uppermost soil layer [Koven et al., 2009]."

*P3, L30: re-order the sentence: "...non-vascular vegetation model LiBry, described in detail by Porada et al. (2013), is integrated into JSBACH."*

We changed this sentence accordingly.

*P4, L2: What is poikilohydry?*

We replaced this term by a description: "..the adaptation of the metabolism to changing levels of water content"

*P5, L3 and P5, L7: These sentences on disturbance appear to be contradicting each other. The first sentence seems to say that the surface coverage is (brutally) set back to a small initial value, while the second one suggests a more gradual, regular disturbance. I think the second sentence is correct. Please clarify.*

The first sentence refers to the original, stand-alone version of LiBry while the second sentence describes the new disturbance scheme of LiBry in JSBACH. We made this clear in the revised manuscript (P. 5, L. 3): ".. a disturbance cycle is included into LiBry which periodically sets back the surface coverage to a small initial value. The surface coverage in steady-state is then obtained by averaging over a whole disturbance cycle.

LiBry in JSBACH is designed to predict the dynamics of the cover in transient scenarios of climate change. Consequently, the steady-state calculation of the surface coverage from the original LiBry is replaced by a dynamic scheme. This also accounts for potential changes in disturbance frequency. To avoid unrealistic fluctuations of the bryophyte and lichen cover, the dynamic disturbance scheme of LiBry in JSBACH accumulates losses of cover over a month. Surface coverage is then updated on a monthly basis: The simulated bryophyte and lichen cover is reduced by the accumulated losses due to disturbance, such as fire, and it is increased by positive net growth accumulated over a month. Net growth is translated into coverage via the specific area of the organisms. .."

*P5, L15: The equation is difficult to read, with all the min and max. It might be useful to clarify right from the beginning that there are one positive growth term, one "negative growth" (senescence?) term and one disturbance term, and to define these separately.*

In the revised manuscript we split the equation and explain the different terms separately (P. 5, L. 16): "..Equation 1 shows the change of bryophyte and lichen cover $\Delta A/\Delta t$ in one month:

$$\Delta A/\Delta t = E - R_{\mathrm{G}} - R_{\mathrm{D}} \tag{1}$$

where $E$ is the expansion of the bryophyte and lichen cover due to positive net growth, which means that accumulated gross photosynthesis exceeds accumulated respiration and biomass turnover. $R_{\mathrm{G}}$ is the reduction of cover due to negative net growth and $R_{\mathrm{D}}$ stands for reduction of cover by disturbance. Expansion of cover $E$ is calculated as:

$$E = \begin{cases} min\,(GA(1.0 - A)\eta_{\mathrm{E}}, 1.0 - A) & \text{if } G \geq 0 \\ 0 & \text{if } G < 0 \end{cases} \tag{2}$$

where $A$ is the surface coverage of bryophytes and lichens in $[\mathrm{m^2\,cover\,m^{-2}\,ground}]$, $G$ is net growth in $[\mathrm{m^2\,new\,cover\,m^{-2}\,cover}]$, accumulated over a month, and $\eta_{\mathrm{E}}$ is a dimensionless "expansion efficiency" which is set to 0.85. This value is taken from the global, stand-alone version of LiBry, which has also been updated to a dynamic disturbance scheme. It was selected to obtain a realistic global distribution of surface coverage. The minimum statement ensures that the cover increment cannot exceed the

available area. The term $A(1.0 - A)$ describes limitation by both existing cover and free area available for growth. Reduction of cover is calculated as:

$$R_G = \begin{cases} 0 & \text{if } G \geq 0 \\ min\,(-GA, A) & \text{if } G < 0 \end{cases} \tag{3}$$

and

$$R_D = DA \tag{4}$$

where $D$ is the fraction of cover lost due to disturbance. It is set to 0.00083 per month. This value corresponds to a fire return interval of 100 years, which is characteristic for the boreal forest [Bonan and Shugart, 1989, Beer et al., 2006, Mouillot and Field, 2005]. The minimum statement in Eq. 3 ensures that the cover reduction cannot lead to negative cover."

*Figure 2: Might be useful to place the values for the "constant" simulation on this graph.*

We added values corresponding to the "constant" model setup to Fig. 2 and we extended the figure caption by the following: "The black dotted lines show the thermal properties of the organic layer from the previous model version [Ekici et al., 2014] for comparison. They are constant since they do not depend on water content."

*P15, L19: It should be said clearly that there is absolutely no reason why you would expect the bryophytes to reduce the cold bias. In the discussion, it would be nice to get an idea about the possible reasons for this substantial bias.*

We added this point to the revised manuscript (P. 12, L. 29): "Comparison of simulated subsoil temperature and active layer thickness to observation-based maps for the region of Yakutia shows that JSBACH still underestimates subsoil temperature, same as previous model versions. This finding is expected since the bryophyte and lichen layer in JSBACH lowers soil temperature and consequently cannot reduce the cold bias."

We also extended the discussion accordingly (P. 14, L. 29): "A probable explanation for the cold bias is the relatively strong sensitivity of annual average soil temperature to the parameterization of the snow layer scheme in JSBACH. Improvements in the representation of snow may improve the simulated soil temperature."

[revised manuscript text omitted]